# Oligodendrocyte precursor cells present antigen and are cytotoxic targets in inflammatory demyelination

Leslie Kirby [1], Jing Jin [1], Jaime Gonzalez Cardona [1], Matthew D. Smith [1], Kyle A. Martin[1], Jingya Wang[2], Hayley Strasburger[1], Leyla Herbst[1], Maya Alexis[1], Jodi Karnell[2], Todd Davidson[2], Ranjan Dutta [3], Joan Goverman[4], Dwight Bergles [5] & Peter A. Calabresi[1,5]

Oligodendrocyte precursor cells (OPCs) are abundant in the adult central nervous system, and have the capacity to regenerate oligodendrocytes and myelin. However, in inflammatory diseases such as multiple sclerosis (MS) remyelination is often incomplete. To investigate how neuroinflammation influences OPCs, we perform in vivo fate-tracing in an inflammatory demyelinating mouse model. Here we report that OPC differentiation is inhibited by both effector T cells and IFNγ overexpression by astrocytes. IFNγ also reduces the absolute number of OPCs and alters remaining OPCs by inducing the immunoproteasome and MHC class I. In vitro, OPCs exposed to IFNγ cross-present antigen to cytotoxic CD8 T cells, resulting in OPC death. In human demyelinated MS brain lesions, but not normal appearing white matter, oligodendroglia exhibit enhanced expression of the immunoproteasome subunit PSMB8. Therefore, OPCs may be co-opted by the immune system in MS to perpetuate the autoimmune response, suggesting that inhibiting immune activation of OPCs may facilitate remyelination.

[1] Department of Neurology, Johns Hopkins School of Medicine, Baltimore, MD, USA. [2] MedImmune LLC, Gaithersburg, MD, USA. [3] Department of Neuroscience, Cleveland Clinic Foundation, Cleveland, OH, USA. [4] Department of Immunology, University of Washington, Seattle, WA, USA. [5] Solomon H. Snyder Department of Neuroscience, Johns Hopkins School of Medicine, Baltimore, MD, USA. Correspondence and requests for materials should be addressed to P.A.C. (email: pcalabr1@jhmi.edu)

Oligodendrocyte precursor cells (OPCs) that express the proteoglycan neuron-glial antigen 2 (NG2) are a highly dynamic, proliferative group of progenitors that remain abundant in the adult CNS. Differentiation of OPCs into oligodendrocytes allows oligodendrocyte generation to continue into adulthood for adaptive myelination and the ability to regenerate myelin following injury or disease. However, the abundance and broad coverage of NG2+ OPCs in both gray and white matter of the CNS, suggests that they may have other functions; indeed, OPCs survey their microenvironment through constant filopodia extension[1–4], migrate to sites of injury and respond to inflammatory cues[5–8], behaviors remarkably similar to microglial cells. The significance of these non-progenitor behaviors in both physiological and pathological conditions are not well understood.

In the setting of chronic inflammatory demyelinating diseases such as multiple sclerosis (MS) the process of endogenous remyelination is inefficient, rendering axons susceptible to degeneration through loss of trophic support and direct toxic effects of immune cells. Recently, subsets of immune cells have been shown to have differing roles in response to demyelination. OPCs express cytokine receptors, and their differentiation is inhibited by interferon-gamma (IFNγ)[9–14], interleukin-17 (IL-17)[5,6], and high doses of tumor necrosis factor (TNF)[15,16], but the mechanistic pathways mediating these effects remain incompletely understood. T regulatory cells (Treg)[7] and alternatively activated monocytes (M2)[17] can facilitate endogenous remyelination through direct actions on OPCs, but Tregs are known to be deficient in MS[18]. Whereas, CD4+ T cells of the Th1 and Th17 lineage have long been recognized to be key in the establishment and perpetuation of MS[19–21]. Although genetic variation in MHC class II alleles has been strongly associated with MS risk implicating CD4+ T cells, CD8+ cells outnumber CD4+ cells in MS lesions[22–24]. In addition, axonal damage closely correlates with CD8+ cell number prevalence rather than CD4 prevalence[25]. While CD4+ and CD8+ cells mediate pathology through cytokine release, the additional cytotoxic properties of CD8+ cells are likely important in the pathogenesis of MS[26–31]. Importantly, recruitment of OPCs to the demyelinated lesion temporally and spatially overlaps with the persistence of CD4+ and CD8+ T cells[32], suggesting that OPCs may be influenced by changes in the microenvironment caused by T cells.

We and others have previously used in vivo genetic fate tracing to track the differentiation of OPCs into mature myelin-producing oligodendrocytes during remyelination in the cuprizone model[33,34]. We demonstrated that adoptive transfer of myelin-reactive T effector cells inhibits endogenous remyelination in the corpus callosum without causing irreparable damage to the axons, as occurs in the spinal cord of animals with experimental autoimmune encephalomyelitis (EAE)[35]. Based on these prior studies we hypothesized that inflammatory cytokines might directly influence the properties of OPCs and inhibit their differentiation.

Here, we identify a signaling pathway in OPCs that both diverts them from differentiating into mature oligodendrocytes and induces expression of antigen-presenting capacity through induction of the immunoproteasome. IFNγ conditioned OPCs highly express MHC class I molecules and present antigens to cytotoxic T cells, in vitro and in vivo. Analysis of postmortem MS tissue reveals the presence of immunoproteasome expressing Sox-10+ cells in white matter lesions, suggesting that OPCs in human disease exhibit similar phenotypic changes. Therapeutic manipulation of this pathway could facilitate functional remyelination by suppressing OPC mediated inflammation, reducing cytotoxic mediated cell death and promoting their differentiation into myelin-forming oligodendrocytes.

## Results

**Effector T cells inhibit remyelination by targeting OPCs.** To better understand the mechanisms by which IFNγ/IL-17 dual producing T cells inhibit remyelination, we adoptively transferred (AT) myelin-reactive cytokine secreting T cells into mice following 4 weeks of 0.2% CPZ[35]. In vivo lineage tracing of OPCs was performed utilizing *PDGFRα-CRE^ER*x*Rosa26-YFP* mice (C57BL/6) (Supplementary Fig. 1a)[33]. The recombined population of OPCs mobilized to promote myelin repair was monitored during remyelination and the influence of the effector T cell transfer was quantified (Supplementary Fig. 1b–e).

We analyzed the differentiation of YFP+ OPCs and myelin content at 1 and 2 weeks after AT (Fig. 1a–d, Supplementary Fig. 2a, b). We detected CD3+ cells at both time points in CPZ and non-CPZ mouse corpus callosum following AT (Fig. 1a, b, Supplementary Fig. 2a, b). Black Gold myelin staining revealed that remyelination was inhibited post AT, but T cells by themselves do not cause demyelination in non-CPZ corpora callosa (Fig. 1b, Supplementary Fig. 2b). A significant reduction in total YFP+ cells was observed at both time points in AT-CPZ mice (Fig. 1c, d). We analyzed the proportion of YFP+ oligodendrocyte lineage cells using the markers PDGFRα, and CC1. In AT-CPZ mice one week post adoptive transfer, the two populations of OPCs (YFP+/PDGFRα+/CC1−) and intermediate oligodendrocytes (YFP+/PDGFRα−/CC1−) were significantly reduced in comparison to CPZ alone (Fig. 1a, c, Supplementary Fig. 2a). The YFP+ mature oligodendrocyte population was significantly reduced in the AT-CPZ mice, as compared to CPZ alone two-weeks post AT (Fig. 1b, d, Supplementary Fig. 2b). Since it would be expected that homeostatic OPC proliferation would maintain cell numbers, an explanation for the reduced number of YFP+ cells might be that the OPCs or oligodendrocyte lineage cells were undergoing cell death specifically in AT-CPZ mice, thus retarding the homeostatic process, a hypothesis that we mechanistically pursued in subsequent experiments.

In order to define the relationship between T-lymphocytes and remyelination, we analyzed the number of CD3+ cells at 1-week and 2-weeks post AT and CPZ withdrawal (Fig. 1e) and found that the density of T cells in the corpus callosum was not significantly different between time points. While MBP staining was reduced in the AT-CPZ mice, the distribution of areas of demyelination and remyelination was variable throughout the corpus callosum of AT-CPZ mice; therefore, we examined the relationship between CD3+ signal intensity and MBP rich or MBP sparse regions (Supplementary Fig. 3a). The signal intensities of MBP and CD3+ were negatively correlated (Supplementary Fig. 3b), supporting the hypothesis that CD3+ cells establish a microenvironment that is not conducive to remyelination. CD4+ and CD8+ T cell densities in the corpus callosum were not significantly different at either time point (Fig. 1f) and both cell types were found to interact equally with YFP+ cells (Fig. 1g). Approximately 20–30% of YFP labeled OPCs were found to be interacting with either CD4+ or CD8+ T cells (Fig. 1h). Taken together, these results suggest that IL-17/IFNγ producing T cells disrupt remyelination independent of the differentiation status of OPCs.

**IFNγ promotes antigen cross-presentation in OPCs.** Since CD4+ T cells exhibit their effector function primarily through cytokine production, we interrogated the influence of IFNγ and IL-17 on the process of OPC differentiation. Primary mouse OPCs were cultured in vitro following A2B5 positive selection (95–98% purity)[36]. OPCs were expanded with PDGF prior to cytokine treatment and differentiation was induced by exposure to T3[37]. Differentiating OPC cultures were treated for a total of

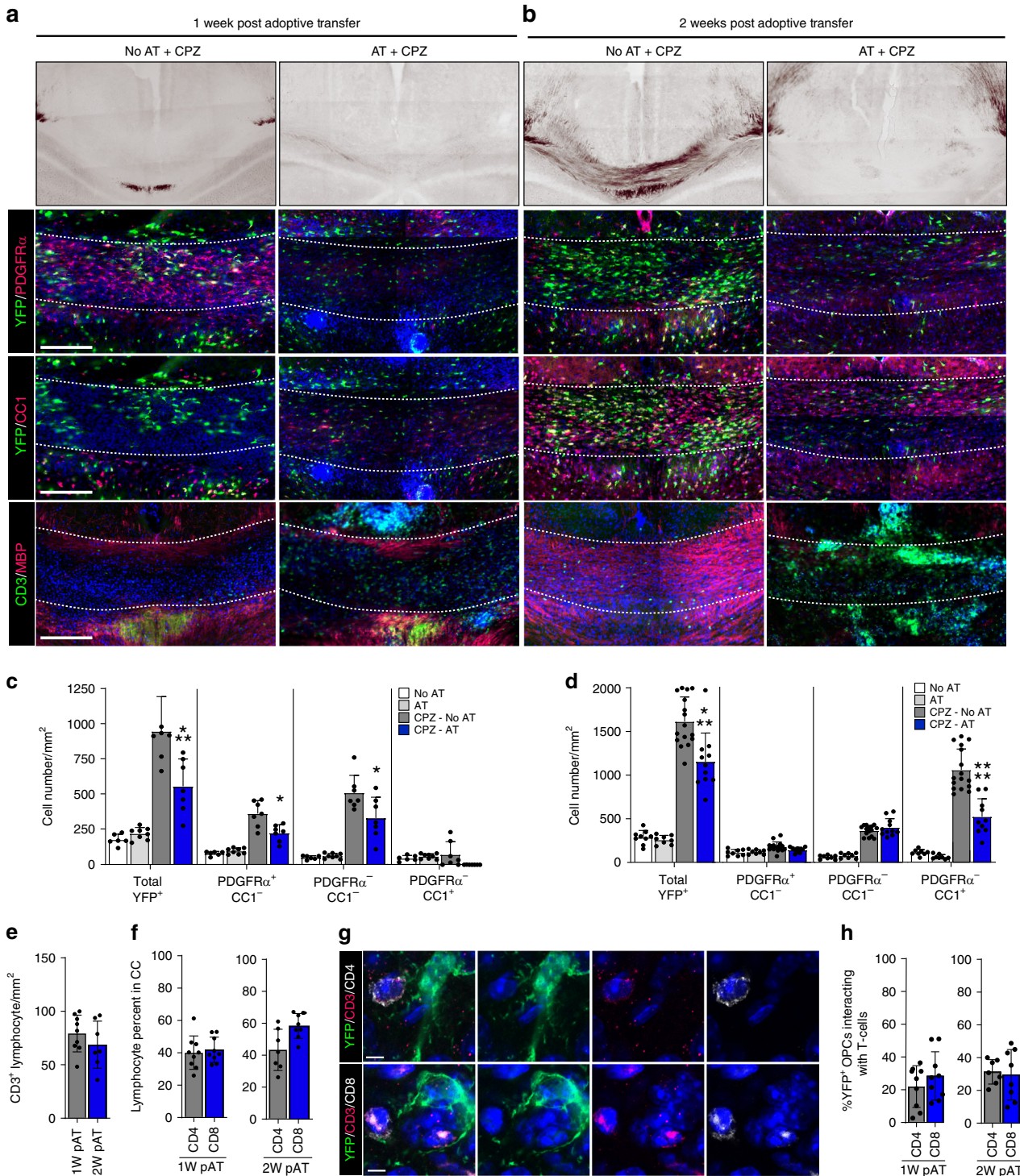

96 hours prior to analysis of gene (Fig. 2a) and protein (Fig. 2b) expression. Cultures exposed to either IL-17 or IFNγ had significantly lower MBP levels, while Sox-10 gene expression and Olig2 protein expression were not significantly altered. We next hypothesized that IFNγ stimulation might have altered the phenotype of OPCs. Microarray analysis of primary OPCs treated with T3 and IFNγ was conducted from 8 to 96 hours of treatment to capture gene expression changes throughout the entire process of OPC differentiation into mature oligodendrocytes (Fig. 2c, d). IFNγ initiated a robust upregulation and downregulation of genes

across multiple time points. Three main gene expression profiles were observed upon comparison with baseline PDGF-treated OPCs (Fig. 2d). Profiles 1 and 3 reveal that IFNγ delayed the influence of T3 on the upregulation or downregulation of gene expression, and profile 2 had a unique pattern, in which gene expression changes were directly linked to IFNγ treatment and independent of the pro-differentiation or pro-progenitor programs.

To interrogate the signaling pathways altered in OPCs by IFNγ, we performed gene set enrichment analysis (GSEA;

**Fig. 1** Effector T cells inhibit remyelination by targeting OPCs. *PDGFRα-Cre^ER x Rosa26-YFP* were kept on a 0.2% CPZ diet for a total of 4-weeks. 4HT injection at 3-weeks allowed for tracking of OPCs through the remyelination process. Approximately 8–10 million $MOG_{35-55}$ specific T cells were injected into recipient mice at 4-weeks. **a** 1-week (scale bar 400 μm) and **b** 2-week images of Black Gold myelin staining (1$^{st}$ row). Representative images of the corpus callosum of brain sections (**a, b** 2nd row–4th row) stained with YFP/PDGFRα (**a, b**-2nd row) allowed tracking of recombined OPCs, stained with YFP/CC1 identified recombined mature oligodendrocytes (**a, b**-3rd row) and stained with CD3/MBP show the distribution of lymphocytes and myelin (**a, b**-4th row). **c, d** Quantification of 1-week (**c**) and 2-week (**d**) immunohistochemistry data to identify different stages of oligodendrocyte differentiation using the markers YFP, PDGFRα, and CC1. Oligodendrocyte lineage populations were compared between groups; No-CPZ (white; $n = 6,8$), No-CPZ+AT (gray; $n = 8,8$), CPZ (charcoal; $n = 7,16$), and CPZ+AT (blue; $n = 7,11$). Significance for quantified data **c, d** was assessed by one-way ANOVA analysis followed by Tukey's multiple comparison analysis ($α = 0.05$, *$\leq 0.05$, **$\leq 0.01$, ***$\leq 0.001$, ****$\leq 0.0001$) [(Total YFP$^+$ mean diff; 5w = 387.2; 95% CI = 153.3−621.1, CC1$^-$/PDGFRα$^+$ mean diff; 5w = 136.8; 95% CI = −28−301.6, CC1$^-$/PDGFRα$^-$ mean diff; 5w = 178.2; 95% CI = 13.39−343), (Total YFP$^+$ mean diff; 6w = 461; 95% CI = 206−716.1, CC1$^+$/PDGFRα$^-$ mean diff; 6w = 535.9; 95% CI = 353.2−718.7)]. **e** Quantification of immunohistochemistry staining of total CD3 in the corpus callosum at 1-week and 2-week post AT (left). **f** Of the total CD3$^+$ percent CD4$^+$ (gray; $n = 9,7$) and CD8$^+$ (blue; $n = 9,8$) was quantified at 1-week (middle) and 2-week (right). **g** Representative confocal images of CD3$^+$/CD4$^+$ T cell interaction (top) and CD3$^+$/CD8$^+$ T cell interaction (bottom) with YFP$^+$ oligodendrocyte lineage cell (scale bar = 2 μm). **h** Proximity analysis of OPC interaction with CD4$^+$/CD8$^+$ T cells at 1 week and 2-weeks post AT. OPC to T cell interaction was determined using Imaris microscopy image analysis software and the percent of YFP$^+$ OPCs interacting with either a CD4$^+$ or CD8$^+$ T cells indicates that OPCs do not preferentially interact with one T cell type at either 1 week or 2-weeks post AT. The range of YFP$^+$ cells interacting with T cells is between 20 and 30% (CD4$^+$ 1-week; 21.9%, 95% CI = 12.2−31.7 CD8$^+$ 1-week; 28.4%, 95% CI = 17.1−39.8, CD4$^+$ 2-week; 31.3% 95% CI = 24.5−38.2, CD8$^+$ 2-week; 25.6%, 95% CI = 16.7−42.4), as analyzed by an unpaired Student's *t*-test ($P$*$\leq 0.05$, **$\leq 0.01$, ***$\leq 0.001$, ****$\leq 0.0001$)

Supplementary Fig. 4a, b). Consistent with the differentiation data, oligodendrocyte markers were more enriched in T3 treatment alone compared to T3 + IFNγ (Supplementary Fig. 4a). Moreover, IFNγ treatment promoted the induction of genes involved in antigen processing and cross-presentation signaling at all time points analyzed (Fig. 2e and Supplementary Fig. 4b; 24 h and 48 h data not shown). Of note, immunoproteasome subunits that define the transition of the proteasome during antigen processing were significantly increased, while constitutive proteasomal subunits were significantly decreased. Furthermore, both MHC class I and class II related genes were significantly augmented in OPCs by IFNγ.

To confirm the microarray data, we performed quantitative PCR analysis and found that mRNAs encoding proteins involved in antigen processing, MHC class I and MHC class II, were significantly increased by IFNγ (Fig. 2f, gray bars). However, the mechanism by which IL-17 orchestrates its inhibitory effect on OPCs is independent of the antigen processing and MHC class I/II signaling, since we were unable to detect upregulation of the genes that comprise these pathways in OPCs that had been exposed to IL-17 (Fig. 2f, blue bars). These results suggest that OPCs undergo phenotypic changes when exposed to IFNγ, adopting features commonly associated with immune cells, such as antigen processing and presentation.

**CNS IFNγ promotes CD8 infiltration and MHC on OPCs.** To determine whether IFNγ promotes OPC antigen presentation in vivo, we first examined whether CNS specific expression of IFNγ was sufficient to upregulate the gene expression profiles that we previously observed. GFAP/tTA transgenic mice were crossed with TRE/IFNγ transgenic mice[12,38]. The Tet-off system allows for controlled IFNγ expression from GFAP$^+$ astrocytes upon removal of dietary doxycycline (DOXY OFF) without the need to induce CNS inflammation, and was previously shown to suppress endogenous remyelination after CPZ[39,40]. Animals were provided 0.2% CPZ to induce demyelination and subsequently increase the pool of OPCs prior to corpus callosum micro-dissection and qPCR analysis of DOXY ON and DOXY OFF mice. Consistent with our in vitro gene expression data, we found that IFNγ induced upregulation of the genes involved in MHC class I and MHC class II antigen presentation (Fig. 3a). To demonstrate the specific effect of IFNγ on OPCs, whole brain tissue was removed after 2-weeks off CPZ and flow cytometry was used to investigate

lymphocytes and OPCs +/−IFNγ in vivo. We compared OPC-H2Kb expression in DOXY ON versus DOXY OFF mice and found significant increases in OPC surface expression of H2Kb in mice expressing CNS IFNγ (Fig. 3b). Upon more detailed analysis of the DOXY OFF H2Kb expressing OPCs we found that a significantly higher proportion of H2Kb$^+$ OPCs were also dual positive for pan MHC class II molecules (IA/IE) than in DOXY ON mice (Fig. 3c). Next, we determined how CNS restricted IFNγ expression influenced T cell and OPC proportions in the brain. Although CD8$^+$ T cells were markedly increased in response to IFNγ, the proportion of CD4$^+$ T cells remained unchanged (Fig. 3d). We interrogated different populations of CD11b$^-$/Olig2$^+$ OPCs based on the three markers: PDGFRα, A2B5, and O4. CNS expression of IFNγ significantly reduced total PDGFRα$^+$, PDGFRα$^+$/A2B5$^+$, and PDGFα$^+$/O4$^+$ expressing OPCs (Fig. 3e). Due to the observed influence on CD8$^+$ T cell percentages, their known cytotoxic effector function, the robust expression of MHC class I and the observed loss of OPCs, we decided to focus on OPC-CD8$^+$ T cell interactions.

**IFNγ induces peptide processing and presentation in OPCs.** To address the functionality of the immunoproteasome and antigen presentation signaling pathways in OPCs, we examined MHC class I (H2Kb in C57/BL6 mice) expression and presentation of MHC class I restricted ovalbumin peptide (OVA$_{257-264}$) following IFNγ stimulation. We treated primary mouse OPCs with IFNγ for 12 h and then provided OVA$_{257-264}$ for 8 h to allow for engulfment and peptide binding in the MHC class I groove, followed by presentation on the cellular surface (Fig. 4a). To identify presentation of OVA$_{257-264}$ peptide on the MHC class I molecule, we utilized an antibody that has specific affinity for OVA$_{257-264}$ loaded H2Kb molecules (H2Kb-OVA)[41]. We observed high expression of H2Kb-OVA in PDGFRα immunoreactive OPCs following IFNγ stimulation, but not with a control MHC class I restricted peptide (MOG$_{37-50}$) or in no cytokine conditions.

MHC class I restricted peptides can be processed and presented through either the cytosolic pathway that is dependent on TAP1 (transporter associated with antigen processing) taking antigen processed by the immunoproteasome and transferring it to the ER, or a vacuolar pathway that is independent of immunoproteasome and TAP1 transport[42–48]. To further explore the dynamics of antigen processing by OPCs, we next determined the

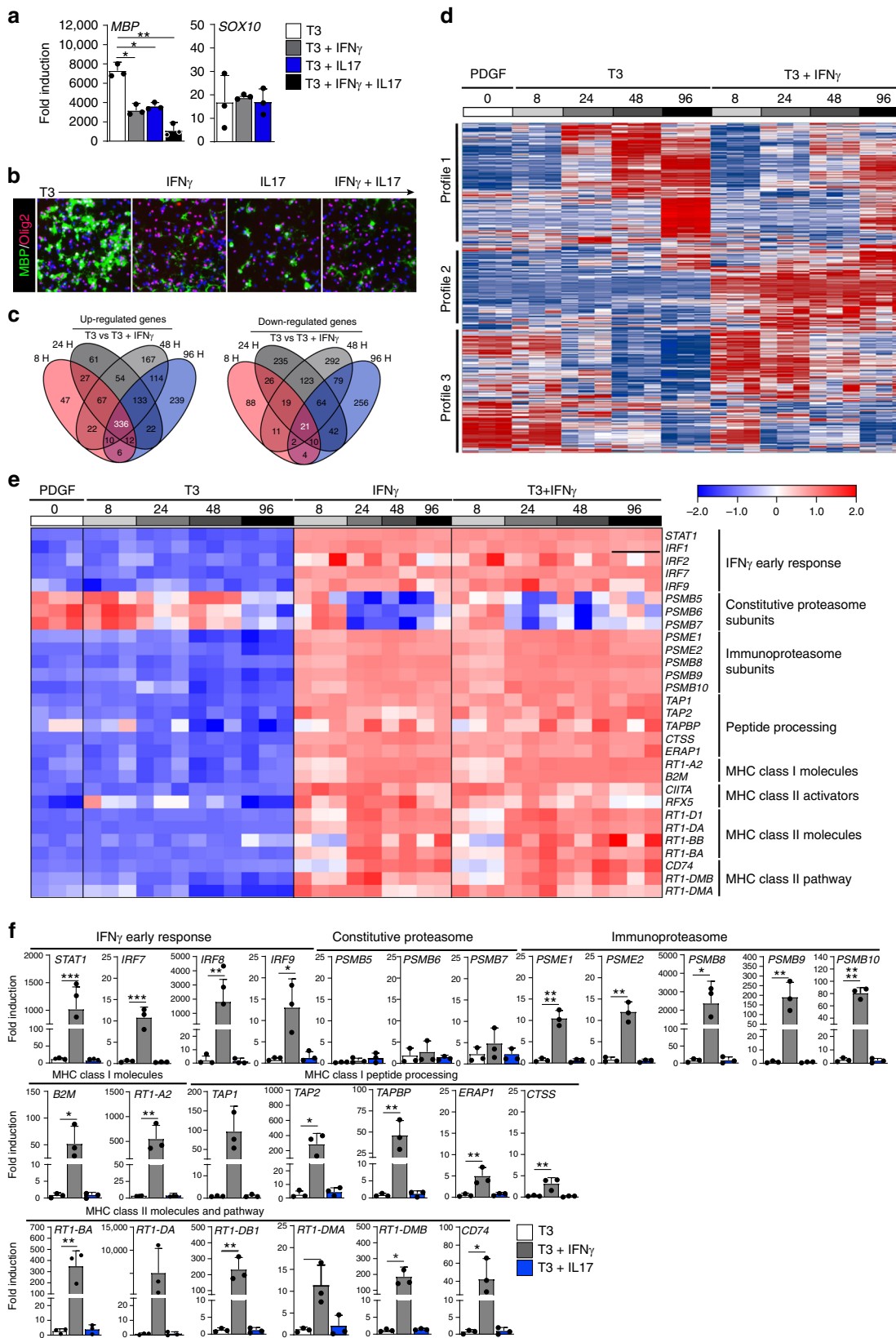

processing/presentation speed and stability of the OVA$_{257-264}$ peptide on MHC class I molecules. After addition of OVA$_{257-264}$ peptide to OPC cultures, we examined the percentage of OPCs presenting H2Kb-OVA (Fig. 4b) and quantified the mean

fluorescence intensity detected (Supplementary Fig. 6b) in WT and TAP1$^{-/-}$ mice. Within 30 minutes of adding OVA$_{257-264}$ to the cultures, over 50% of the IFNγ stimulated OPCs presented the peptide. Although the percentage of OPCs expressing H2Kb-

**Fig. 2** IFNγ promotes antigen cross-presentation in OPCs. Isolated and PDGF (20 ng/mL) expanded postnatal rat OPCs (P4-P6) were differentiated with T3 (white), T3+IFNγ (10 ng/mL; charcoal), T3+IL17 (50 ng/mL; blue), or T3+IFNγ+IL-17 (black) for 96 h prior to assessment using qPCR (**a**) and ICC staining for MBP and Olig2 (**b**). Isolated and PDGF (20 ng/mL) expanded postnatal rat OPCs (P4-P6) were left undifferentiated (0 h; white) or differentiated with T3 (10 nM), IFNγ (10 ng/mL), T3 + IFNγ, for 8 hrs (light gray), 24 hrs (medium gray), 48 h (charcoal) and 96 hrs (black) to compare gene expression between groups while obtaining information at multiple time points. **c–e** Affymetrix gene arrays analysis was performed from three independent biological replicates. **c** Venn diagram summarizing the number and overlap of up-regulated (left) and down-regulated (right) genes based on time point. **d** Global heat map of all probes clustered into three gene expression patterns comparing PDGF, T3, and T3+IFNγ at all time points. **e** Targeted heat map comparing PDGF, T3, IFNγ, and T3+IFNγ at all time points. Displayed genes were identified from GSEA analysis (supplement 3). **f** Quantitative PCR validation of OPCs differentiated with T3 (white), T3+IFNγ (10 ng/mL; gray), or T3+IL17 (50 ng/mL; blue) for 96 h prior to assessment. Error bars represent the standard deviation from three independent primary isolations and experiments. Significance for qPCR analysis was determined by one-way ANOVA analysis followed by Dunnett's multiple comparison analysis where T3+IFNγ or T3+IL17 were compared to T3 alone control ($P^* \leq 0.05$, $^{**} \leq 0.01$, $^{***} \leq 0.001$, $^{****} \leq 0.0001$).

OVA remained stable at 80–90% between 2 and 64 h, the mean fluorescence intensity continued to increase through 32 h. The dynamics of OPC peptide presentation were comparable to bone marrow derived dendritic cells (DCs)[49]. One caveat of this experiment is that $OVA_{257-264}$ peptide does not require cytosolic MHC class I processing, as shown by the ability of $TAP1^{-/-}$ OPCs to express surface H2Kb-OVA (Fig. 4b, right, Supplementary Fig. 5a). However, given that MHC class I is only expressed on the surface after intracellular peptide loading, it is unlikely that OVA peptide was externally coating MHC class I molecules, because H2Kb is detected on the surface of $TAP1^{-/-}$ OPCs only after $OVA_{257-264}$ is provided. Therefore, these results suggest that OPCs engulfed the peptide and internally loaded it onto MHC class I molecules through a $TAP1^{-/-}$ independent mechanism termed the vacuolar pathway, as has been described[48]. We subsequently verified the distinct processing of whole OVA protein as being dependent on the cytosolic pathway (Fig. 7c). Since oligodendrocytes have previously been shown to present antigen, we compared the capacity of IFNγ stimulated OPCs, intermediate oligodendrocytes, and mature oligodendrocytes to present MHC class I[50]. OPCs were approximately 85.3% MHC class I^hi after 24 h of IFNγ stimulation compared to intermediate (54.3%) and mature oligodendrocytes (31.2%) (Fig. 4c). Notably, OPCs that express MHC class I under IFNγ stimulation also express MHC class II (Supplementary Fig. 6), suggesting that OPCs may also activate CD4 cells.

**OPCs cross-present ovalbumin and activate CD8⁺ T cells**. To further explore the mechanisms that support OPC antigen presentation, we developed an OPC-CD8 co-culture system. OPCs were treated with IFNγ for 12 h prior to OVA protein administration. Use of the full-length OVA protein necessitates engulfment, immunoproteasome processing, and TAP1/2-dependent ER transport to successfully achieve surface expression of OVA peptide on MHC class I. OT-1, $OVA_{257-264}$ peptide MHC class I restricted TCR transgenic mice were utilized to examine CD8 activation[51,52] (Fig. 5a). At 24 and 48 h after co-culture initiation, CD8⁺ cell (Fig. 5b, top left) activation was examined by morphology, survival, CD25/CD69 immunoreactivity, and proliferation (Fig. 5b). Consistent with an activated phenotype, only OT-1 CD8s co-cultured with IFNγ stimulated+ovalbumin OPCs appeared clustered under phase contrast and had significantly higher survival, percentage of CD25⁺/CD69⁺ cells, and cell proliferation at 48 hours compared to controls. Importantly, OT-1 CD8s co-cultured with IFNγ/ovalbumin OPCs were similar in their ability to survive and proliferate, when compared to splenocytes presenting processed OVA (Supplementary. Fig. 7a).

Complete CD8 cytotoxic T cell (CTL) activation results in cytokine and granular protein production, providing CD8s with their effector function[26–28]. We investigated the ability of IFNγ-

ovalbumin OPCs to induce OT-1 CTLs and found that significantly more CD8s were induced to produce IFNγ, TNFα, perforin, and granzyme B when compared to the other experimental conditions (Fig. 5c). The OPC induced CD8 cytokine production was comparable to that of splenocyte (containing classical antigen presenting cells) induced CD8 cytokine production of IFNγ and TNFα (Supplementary Fig. 7b). Together, these experiments indicate that IFNγ stimulated OPCs engage in antigen cross-presentation through the engulfment, processing, and surface presentation of antigen, resulting in the activation of CD8⁺ CTLs.

**Caspase3/7 increases in OPCs that activate OT-1 CD8⁺ T cells**. CD8⁺ T cell cytotoxic targeted cell death is a potential mechanism for remyelination failure in MS if OPCs are targeted. Both protease-mediated cytotoxicity and contact-dependent Fas-Fas Ligand surface interaction allow CD8⁺ T cells to induce apoptosis of its target while minimizing off-target effects[29,30]. Although perforin/granzyme and Fas/FasL intracellular signaling can be divergent, both pathways have been shown to promote apoptosis through caspase 3/7 activation[53–55]. To determine whether OPC-activated CD8⁺ cells were in turn cytotoxic, we first examined if OPCs exhibited surface expression of Fas by flow cytometry (Fig. 6a). A significantly higher percentage of OPCs expressed surface Fas after IFNγ treatment compared to non-stimulated OPCs. We used live cell imaging of OPC-CD8⁺ cell co-cultures to measure induced caspase3/7 activity of OPCs (Fig. 6b–h). OT-1 CD8⁺ cells and OPCs cultured in isolation exhibited very little caspase 3/7 activity (Fig. 6c). Non-stimulated OPCs cultured with or without antigen at baseline, 24 h, and 48 h had similar low levels of caspase 3/7 as IFNγ stimulated OPCs without antigen (Fig. 6e; white mask). IFNγ+ovalbumin conditioned OPCs that robustly activated OT-1 CD8⁺ cells had significantly more cell death when compared to IFNγ/no antigen OPCs (Fig. 6e). When analyzing the decrease in NucLight labeled OPCs there was a significant decrease in total number between 36–48 hours only in OPC cultures that were pre-treated with IFNγ+ovalbumin prior to CD8⁺ T cell addition (Fig. 6f). Specific blockade of granzyme B and FasL resulted in significantly less cell death when compared to vehicle treated OPCs (Fig. 6g). Furthermore, there was not a difference between the inhibitor treated cultures and the culture treated with a pan caspase inhibitor (Fig. 6h; right).

To examine the effects of antigen concentration on CD8⁺ T cell cytotoxicity, we performed serial dilutions of $OVA_{257-264}$ and ovalbumin, and found significantly inhibited caspase 3/7 activity at lower compared to higher concentrations (Fig. 7b). Pharmacological blockade of different stages of the cytosolic MHC class I pathway using; Chloroquine, ONX-0914 (PSMB8 inhibitor), or Cathepsin S inhibitor had no effect on OPC-$OVA_{257-264}$ peptide

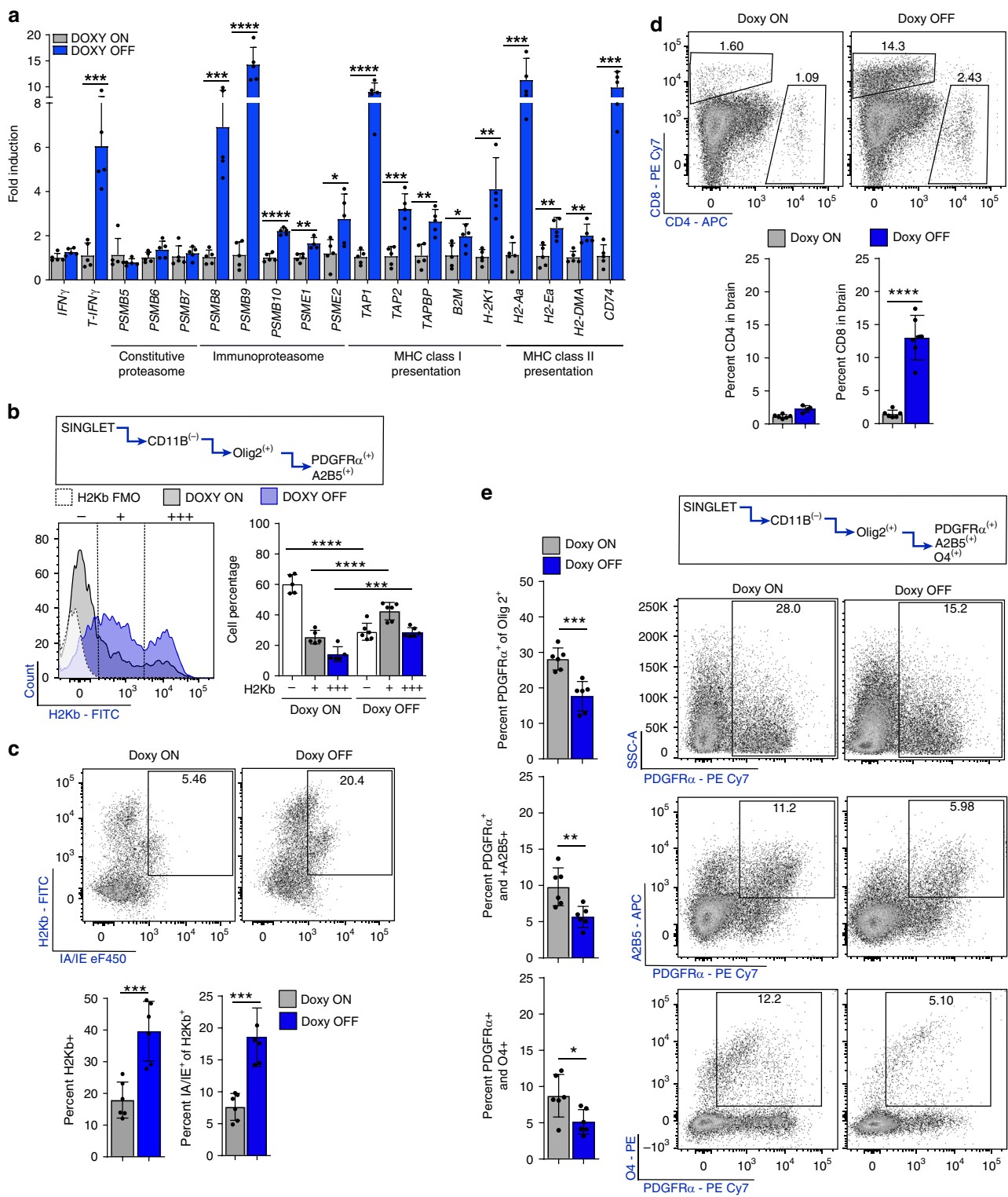

presentation, as determined by the OT-1 CD8 proliferation assay, while all three significantly hindered the ability of OPCs to engulf, process and present ovalbumin protein. These results are consistent with the hypothesis that peptides can be processed through the above described alternative vacuolar pathway, but whole protein requires the classical cytosolic processing machinery (Fig. 7c). Furthermore, the antigen pathway processing blockers had similar effects as antigen concentration dilution in which the caspase 3/7 activity curve was shifted to the right by

approximately 25 hours (Fig. 7d, e). These experiments confirmed reciprocal effects of the activated cytotoxic CD8$^+$ cells on OPCs and underline two mechanistic pathways leading to cell death that could account, in part, for the observed OPC depletion in vivo (Figs. 1a–d, 3d, e).

**OPCs present myelin peptides for cytotoxicity in vivo.** To examine the consequences of MHC class I antigen presentation

**Fig. 3** CNS restricted IFNγ promotes CD8 infiltration and MHC on OPCs. *TRE/IFNγ × GFAP/tTA* mice were fed CPZ for 6 weeks and either maintained on doxycycline (gray; $n = 5$) or removed from doxycycline (blue; $n = 5$) to induce IFNγ expression. Two-weeks after CPZ withdrawal the corpus callosum was dissected and qPCR analysis was performed (a). Significance for qPCR was determined by two-tailed, unpaired *t*-test between doxycycline and no doxycycline conditions. **b** Whole brains from *TRE/IFNγ × GFAP/tTA* mice under the same experimental paradigm were isolated for flow cytometric analysis. The OPC population was determined by CD11b negativity and Olig2, A2B5, and PDGFRα positivity. H2Kb expression is shown in the histogram plot in which the staining control FMO (white) is compared to mice kept on doxycycline (gray; $n = 5$) and mice removed from doxycycline (blue; $n = 5$). Significance for flow cytometry was determined by two-tailed, unpaired Student's *t*-test between doxycycline and no doxycycline conditions (H2Kb$^-$ mean diff $= -31.3 \pm 3.5$; 95% CI $= 39.3-(-)23.4$, H2Kb$^+$ mean diff $= 17.0 \pm 3.2$, 95% CI $= 9.9-24.2$, H2Kb$^{+++}$ mean diff $= 14.3 \pm 2.4$, 95% CI $= 8.8-19.7$). **c** The same gating strategy was used to determine MHC Class II (IA/IE) expression by H2Kb. Significance for flow cytometry was determined by two-tailed, unpaired *t*-test between doxycycline (gray; $n = 6$) and no doxycycline (blue; $n = 6$) conditions were compared (IAIE$^+$ of H2Kb$^+$ mean diff $= 10.9 \pm 2.1$, 95% CI $= 6.3-15.5$). CD4$^+$/CD8$^+$ populations were also analyzed within the whole brain tissue as determined by flow cytometry. The CD4$^+$ population remains unchanged between doxycycline (gray; $n = 6$) and doxycycline removal (blue; $n = 6$). However, the removal of doxycycline significantly increased the number of CD8$^+$ cells in the brain. Significance for flow cytometry was determined by two-tailed, unpaired Student's *t*-test between doxycycline and no doxycycline conditions were compared, (CD8$^+$ T cell mean diff $= 11.55 \pm 1.399$, 95% CI $= 8.429-14.66$). **e** Percent OPC numbers were also analyzed for PDGFRα, A2B5, and O4. PDGFRα$^+$ (top), PDGFRα$^+$/A2B5$^+$ (middle), and PDGFRα$^+$/O4$^+$ (bottom) OPC percentages were all significantly decreased in animals in which doxycycline was removed. Significance for flow cytometry was determined by two-tailed, unpaired Student's *t*-test between doxycycline and no doxycycline conditions were compared (PDGFRα$^+$ mean diff $= -10.5 \pm 2.1$, 95% CI $= -15.3-(-)5.8$, PDGFRα$^+$/A2B5$^+$ mean diff $= -4.1 \pm 1.2$, 95% CI $= -6.9-(-)1.4$, PDGFRα$^+$/O4$^+$ mean diff $= -3.6 \pm 1.4$, 95% CI $= -6.7-(-)0.5$). All error bars represent standard deviation. ($P^*\leq0.05, ^{**}\leq0.01, ^{***}\leq0.001, ^{****}\leq0.0001$)

in vivo, we utilized two different strains of mice, each with unique advantages. First, we adoptively transferred 2D2[56] MOG$_{35-55}$-specific T cells into the *PDGFRα-CRE$^{ER}$ × Rosa26-YFP* cuprizone fed mice, which enabled us to trace OPCs with a reporter and have greater certainty of OPC-specific MHC class I expression and CD8$^+$ cell CTL mediated cytotoxicity (Supplementary Fig. 8a). This model also has a detectable CD8$^+$ cell infiltrate even though disease is induced with CD4$^+$ cells[35]. We found significantly higher EAE scores in AT-CPZ-mice compared to no-AT-CPZ (Supplementary Fig. 8b). In addition, these mice showed pronounced and significantly more CD3$^+$ lymphocytes (Supplementary Fig. 8c). H2Kb and Fas expression within the total OPC population was significantly higher in the mice with AT versus no-AT (Supplementary Fig. 9a). Using flow cytometry, we gated on the YFP$^+$/PDGFRα$^+$/A2B5$^+$ population in AT-CPZ and no AT-CPZ control and found a significant proportion of both YFP$^+$ and YFP$^-$ OPCs that were caspase 3/7 active (Fig. 8a). Of note, there were significantly fewer YFP$^+$ OPCs in mice that received the AT, as determined by flow cytometry, making this result consistent with previously shown immunohistochemical analysis (Fig. 8a). We extended the analysis of this population and interrogated cell death, H2Kb expression, and Fas expression using flow cytometry. The majority of the YFP$^+$/caspase 3/7 active population were labeled dead, H2Kb positive and Fas positive (Supplementary Fig. 9b).

We utilized an additional adoptive transfer model to further validate OPC antigen presentation and to measure MBP$_{79-87}$ peptide on MHC class I using an antibody specific for this peptide on H2Kk in C3HeB/FeJ mice[50,57]. This system was previously employed to show that TNF/iNOS producing DCs (Tip-DCs) and mature oligodendrocytes mediate determinant spreading in EAE (Supplementary Figs. 10a–d and 11a–c)[50]. Examination of H2Kk and Fas expression were consistent with the previous model in which the AT with and without CPZ fostered increased expression of both markers in OPCs (PDGFRα$^+$/A2B5$^+$/CD11b$^-$/CD45$^-$) (Fig. 8b). In MOG$_{97-114}$ AT mice there were a significantly higher proportion of OPCs with induced caspase 3/7 activity (Fig. 8c). When comparing AT-caspase 3/7 active OPCs with CPZ-AT-caspase3/7 active OPCs, we found there was no difference in the percentage of dead cells, but there was a significantly higher proportion of H2Kk and Fas immunoreactive cells in mice fed CPZ (Fig. 8c). One possible explanation for this difference could be the level of IFNγ production by lymphocytes

between AT alone and AT-CPZ. In accordance, we previously found that CPZ shifts the lymphocyte brain infiltrate from higher percentages of IL-17 producing CD4$^+$ cells to predominantly dual producing IFNγ/IL-17 CD4$^+$ cells[35]. Using an antibody specific for H2Kk-MBP$_{79-87}$, we next found increased MBP$_{79-87}$ MHC class I restricted peptide expression on OPCs in animals following AT-CPZ vs CPZ alone (Fig. 8d), suggesting that under inflammatory conditions, OPCs could contribute to determinant spreading. Together, these data show that in an inflammatory setting, OPCs can be induced to function as antigen presenting cells in vivo and present myelin peptide on MHC class I molecules. Moreover, at peak disease, a significant percentage of OPCs are targeted to undergo caspase 3/7 mediated death, providing a mechanistic explanation for the lower density of OPCs observed following oligodendrocyte depletion. We also identified granzyme B$^+$ target cells in the inflamed brain after AT (Fig. 9b) amongst non-lymphocyte cell populations that were granzyme B$^+$ the CD45hi/PDGFRα$^-$/CD11c$^-$ DCs, CD45$^{lo}$/PDGFRα$^-$/CD11b$^+$ microglia, and PDGFRα$^+$/A2B5$^+$ OPCs (Fig. 9c). Upon further analysis of these granzyme B$^+$ populations we found that OPCs specifically were both caspase 3/7 active and not viable, whereas DCs were granzyme B$^+$ and viable and microglia were granzyme B$^+$ and caspase$^+$ and viable (Fig. 9e). These results are consistent with prior studies showing that granzyme B activation in DCs is inhibited by the serpin proteinase inhibitor 9[58], and caspase-3/7 activation occurs in microglia without triggering cell death in vitro and in vivo[59], but suggest that OPCs are a more vulnerable population to immune cell cytotoxicity at peak disease when compared to microglia and DCs.

**MS lesions have high PSMB8 expression in Sox-10$^+$ cells.** To expand our studies of OPC phenotypic changes to human tissue samples, we conducted immune-histochemical (IHC) staining of human brain tissues collected from control and MS patients through autopsy. Using myelin proteolipid protein (PLP) staining, regions of normal white matter (NAWM) and white matter lesions (WML) were identified. Staining for PSMB8, a subunit specific to the immunoproteasome, was used to define cells that had activation of the cross-presentation pathway (Fig. 10a). Highly intense and prevalent staining for PSMB8 was observed in WML areas compared to both control and NAWM tissues. Furthermore, there were significantly more cells that were

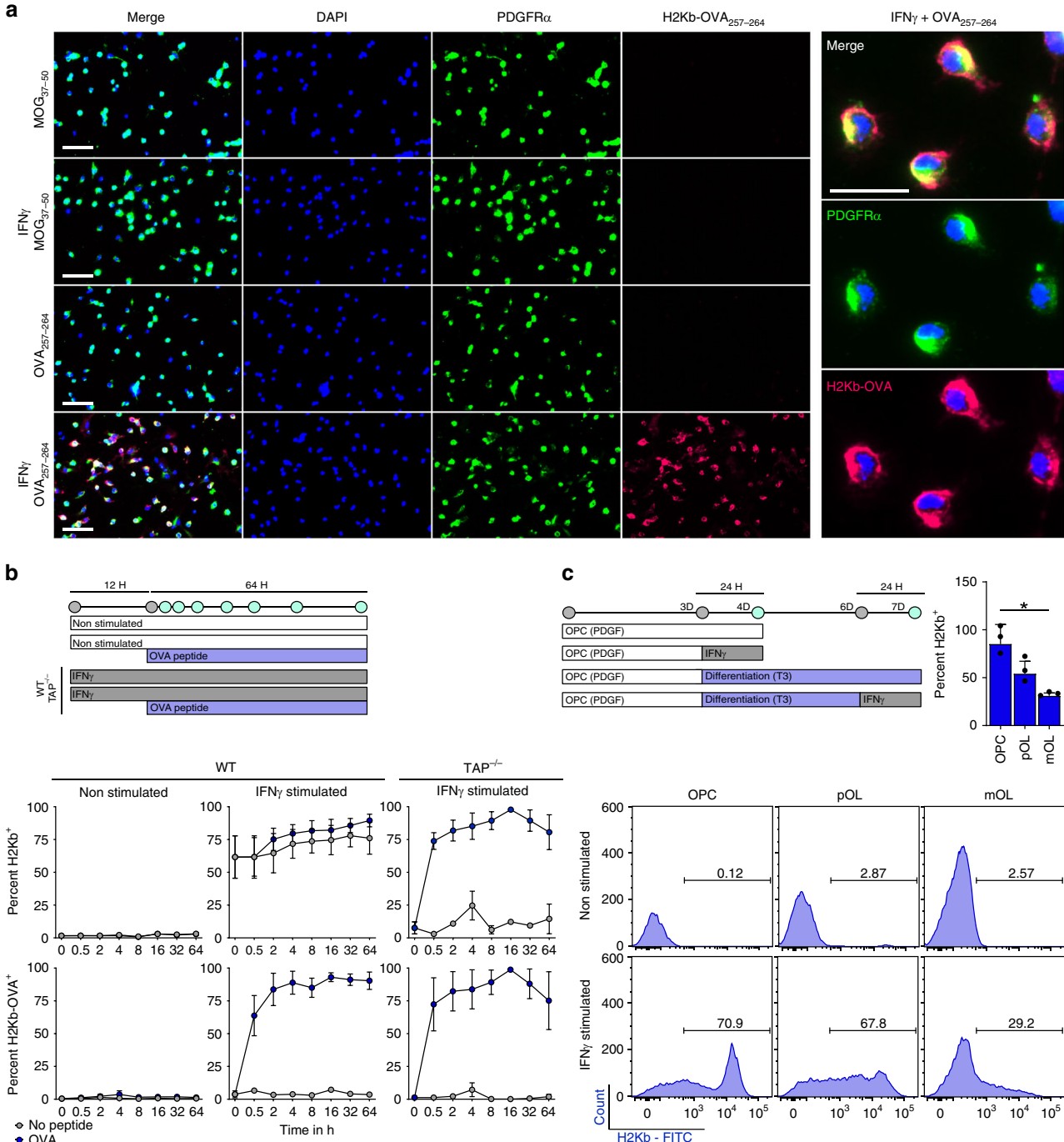

**Fig. 4** IFNγ induces peptide processing and presentation in OPCs. **a** Immunocytochemical staining of primary C57BL/6 mouse OPCs which were cultured under PDGF conditions (20 ng/mL) or with IFNγ (10 ng/mL) for 12 h prior to addition of the MHC class I restricted MOG$_{37–50}$ peptide (50 μg/mL) or OVA$_{257–264}$ peptide (50 μg/mL) (scale bar = 50 μm). MCH class I restricted ovalbumin peptide presentation on OPCs was determined using PDGFRα and H2Kb-OVA$_{257–264}$ antibodies. The right panel shows high magnification (25 μm) of IFNγ + OVA$_{257–264}$ culture. **b** Time course experiments of WT and TAP$^{−/−}$ OPCs cultured with/without IFNγ and with/without OVA$_{257–264}$ peptide. Treatment of IFNγ was completed for 12 h prior to the addition of no peptide (gray) or OVA$_{257–264}$ peptide (blue) (**c**). MHC class I expression was determined from three stages of oligodendrocyte lineage cells either unstimulated or stimulated with IFNγ for 24 h. Oligodendrocyte lineage populations were defined by PDGFRα, A2B5, and O4 markers under proliferative or differentiating conditions (OPC vs. pOL mean diff = 30.9, 95% CI = −13.6–75.5, OPC vs. mOL mean diff = 54.1, 95% CI = 5.2–102.9)

PSMB8$^+$ in the WML regions compared to NAWM and healthy control brain white matter (Fig. 10b). To ascertain which cells within the lesion were expressing PSMB8, parallel sections were co-immunostained with the oligodendrocyte lineage marker Sox-10 (Fig. 10c). There were significantly more PSMB8$^+$/Sox-10$^+$ cells in WML compared to control and NAWM (Fig. 10d), indicating that oligodendrocyte lineage cells within human white matter MS lesions upregulate a key component of the immuno-proteosome pathway. Taken together, the results show that oligodendrocyte lineage cells in MS WML upregulate the immunoproteosome pathway specifically in areas of failed remyelination.

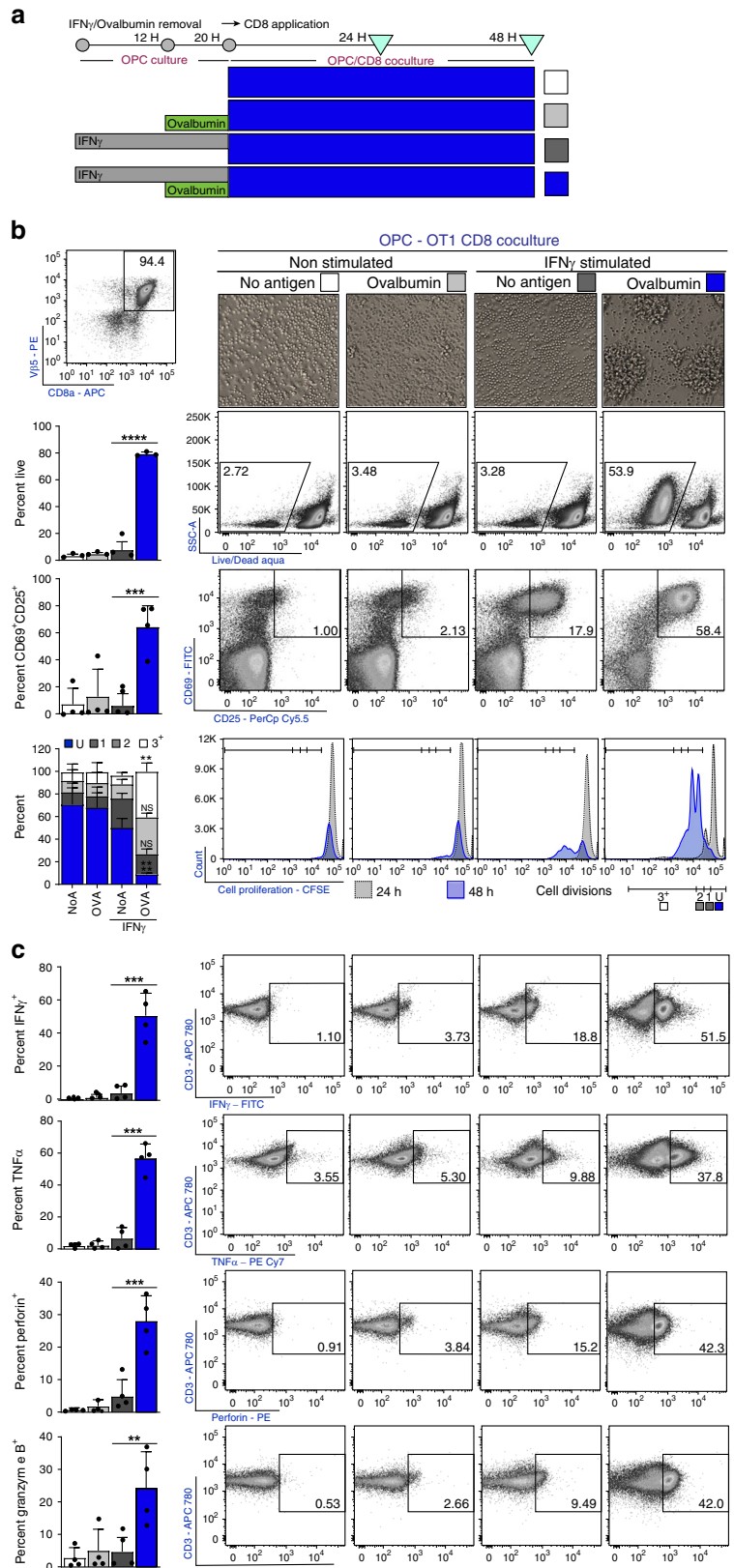

## Discussion

Our findings demonstrate that OPCs fail to differentiate into mature oligodendrocytes in an inflammatory demyelinating mouse model and are capable of becoming antigen presenting cells. IFNγ induces expression of the MHC class I antigen presentation pathway in OPCs, and in vitro, confers an ability to activate CD8+ T cells, which can in turn kill the OPCs as target cells. OPCs can cross present exogenous proteins on

**Fig. 5** OPCs cross-present ovalbumin and activate CD8[+] T cells. **a** Timeline and experimental design. OPCs were cultured in PDGF to inhibit differentiation and stimulated with IFNγ (10 ng/mL) for 12 h prior to Ovalbumin (500 μg/mL) protein addition. Both IFNγ and Ovalbumin protein were incubated with OPCs for a total of 8 h prior to washing the cultures to remove unprocessed Ovalbumin. OT-1 CD8[+] T cells were isolated by magnetic sorting then stained with Cell Proliferation Dye eFluor 450 (10 μM) prior to initiation of CD8/OPC co-culture. In all, 24–48 h after the start of the co-culture CD8 were analyzed for activation. (**b**; top left) CD8 percentage, at 24 h after the co-culture was initiated, was determined using Vβ5, CD3, and CD8, and subsequently used as a parent gate for all flow plots in the figure. **b** CD8 morphology (top), survival (2nd row) (mean diff = −71.6, 95% CI = −107.7 to −35.6), activation status using CD25[+]/CD69[+] (3rd row) (mean diff = −58.1, 95% CI = −32.7 to −6.6), and proliferation (bottom) (Undivided mean diff = 41.1, 95% CI = 17.8–64.4, 3[+] Division mean diff = −32.8, 95% CI = −56.1 to −9.5) with quantification (left to each panel) of OT-1s cultured with OPCs; no peptide (white; n = 4), Ovalbumin (light gray; n = 4), IFNγ + no peptide (gray; n = 4), and IFNγ + Ovalbumin (blue; n = 4). **c** Cytokine and granular protein profiling of OT-1s; IFNγ (1st row) (mean diff = −46.5, 95% CI = −73.4 to −19.36), TNFα (2nd row) (mean diff = −50.1, 95% CI = −72.9 to −27.3), perforin (3rd row) (mean diff = −23.2, 95% CI = −35.6−(−)10.8) and granzyme B (4th row) (mean diff = −32.7, 95% CI = −32.7−(−)6.6). Significance for all quantified data was assessed by one-way ANOVA analysis followed by Tukey's multiple comparison analysis (P*≤0.05, **≤0.01, ***≤0.001, ****≤0.0001). Error bars represent standard deviation for four biological replicates

class I molecules, a process that has previously been attributed to subsets of professional antigen presenting cells such as DCs[47,60]. We demonstrate that OPCs switch from the constitutive proteasome to the immunoproteasome when exposed to IFNγ. In postmortem MS brain WML, we observed marked upregulation of the immunoproteasome subunit PSMB8 on oligodendrocyte lineage cells, suggesting this process may be a critical component of the chronically demyelinated lesions of people with longstanding MS. Therefore, we suggest that OPCs in an inflamed CNS may not only fail to facilitate remyelination, but could actually propagate chronic inflammation. These data provide mechanistic insight to several recent reports showing that OPCs and oligodendrocyte lineage cells from EAE and people with MS express transcripts associated with inflammation and antigen presentation[61,62].

MS pathological studies document CNS CD8[+] cell activation and clonal CD8 expansion to cognate antigen of cells localized to the parenchyma has been observed[23,24,63]. Furthermore, CD8[+] T cells outnumber CD4[+] T cells in MS brain lesions that are also populated by recruited OPCs[64,65]. Our results, showing that OPCs can cross present antigen on MHC class I molecules and activate CD8[+] T cells, may explain the longstanding pathological observations of CD8[+] cell predominance in the MS lesion. OPCs may have this signaling pathway in order to participate in responses to CNS infections. The subsequent OPC death may be an acceptable resolution to acute infection/inflammation given their abundance in the CNS, as a means to remove cells that are dedicated to this immune-modulatory function, much in the way DCs in the periphery die after presenting antigen.

A prior report from one of us (JG) documented that Tip-DCs cross present antigens in EAE[50]. This study also identified that mature oligodendrocytes, but not microglia, were able to mediate determinant spreading and activation of CD8[+] T cells. Here we show OPCs express higher levels of peptide loaded class I molecules than do mature oligodendrocytes. Importantly, we were able to demonstrate that OPCs present MBP[79–87] MHC class I restricted peptide in vivo, but similar to the previous study, were unable to detect this peptide loaded MHC class I molecule expression within the microglial population. We also observed upregulation of MHC class II and several co-stimulatory molecules in our microarray and further studies need to be performed to further interrogate this pathway since examination of the ability of OPCs to activate CD4[+] T cells is clearly of relevance to MS.

In postmortem MS plaques, we saw a substantial proportion of oligodendrocyte lineage cells were present and had high expression of immunoproteosome subunit, PSMB8, but only in the areas of failed remyelination and not the normal appearing white matter. This finding is consistent with the notion that OPCs were

either arrested in their maturation and or have taken on an alternative role to present antigen. Since mature oligodendrocytes from MS tissue have been shown to be immunoreactive for Fas surface expression, we propose that OPCs also may be targeted for Fas mediated cells death in MS[66]. We have strong evidence for Fas/FasL signaling contribution to cytotoxicity in vitro and future studies will investigate this pathway in vivo. Furthermore, perforin and granzyme signaling is also important for CD8[+] CTLs and natural killer cell cytotoxicity. In a recent meta-analysis of 120,991 low-frequency coding variants, the International MS Consortium of Genetics (IMSCG) identified that a variant in *PRF1*, the gene that encodes perforin, is associated with the risk of acquiring MS[67]. We provide in vitro and in vivo data that supports the role of granzyme B induced cytotoxicity. Thus, not only do the OPCs fail to differentiate, but they are co-opted by the immune system to propagate the CNS immune response.

An important area of future study will be to understand the relative contribution of OPC antigen presentation as compared to other classical CNS APCs in demyelinating diseases. One intriguing result we found from our study is that even though we were able to identify granzyme B positive DCs, microglia and OPCs, only granzyme B[+] OPCs were caspase 3/7 active and were dead. This result indicates that OPCs may be more vulnerable than other antigen presenting cells. In fact, the granzyme B[+] microglia were caspase 3/7 active but alive while granzyme B[+] DCs were not caspase 3/7 active and were alive. These two populations have been previously shown to have important signaling mechanisms for utilization of caspase activity and protection from granzyme B induced apoptosis, respectively[58,65]. Caspase activity in microglia is necessary for activation and iNOS production as the caspases are utilized to induce the NF-kB pathway. DCs express serpine molecules that target and cleave intracellular granzyme B and arrest the apoptotic signal upstream of caspase activation.

If this process were operational in MS, as suggested by our observation of marked expression of PSMB8 only in areas of MS brain demyelination, it could suggest that either depleting or redirecting the transcriptional program of pathogenic OPCs may be beneficial for myelin repair. Targeting these newly discovered inflammatory signaling pathways in OPCs may be an important step in blocking inflammatory responses of OPCs, and to facilitate therapies designed to promote myelin repair, which to date have primarily targeted developmental pathways. While proteasomal inhibitors are used in malignancies such as myeloma, they are quite toxic. Nevertheless, targeting one of the three critical subunits of the immunoproteasome (PSMB8, 9, and 10) might allow selective suppression of antigen cross presentation without suppressing normal cell function[68]. Furthermore, enhancing signaling mechanisms that reduce the OPC vulnerability to cytotoxicity, should be a major focus for future studies.

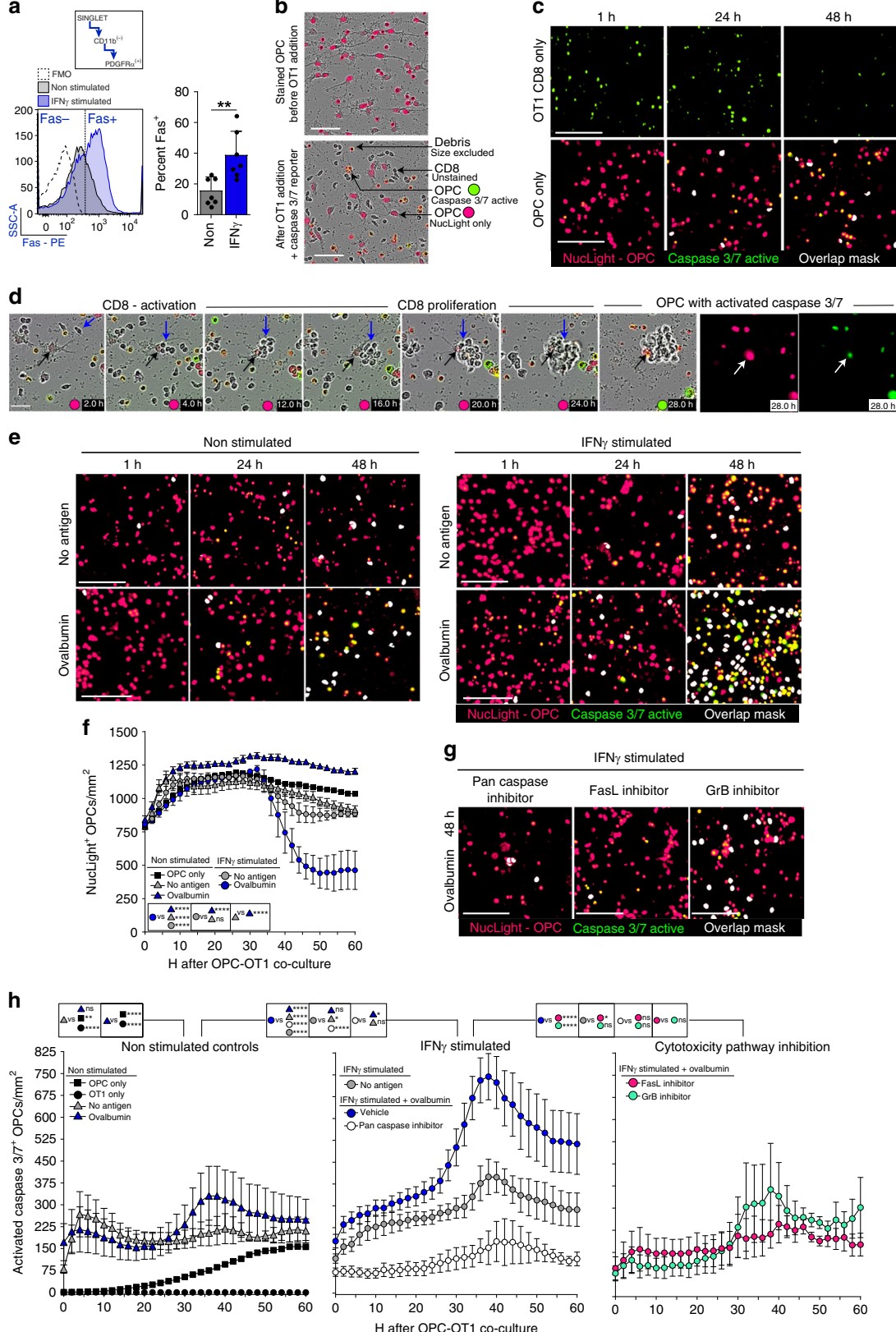

## Methods

**Mice**. Mice were housed and maintained in a pathogen-free animal facility at Johns Hopkins University. *TRE/IFNγ* and *GFAP/tTA* mice were a kindly provided by B. Popko (University of Chicago) and were then crossed and maintained in-house. OVA$_{257-264}$ TCR transgenic, OT-1 mice (C57BL/6-Tg(TcraTcrb)1100Mjb/J), and

2D2 MOG$_{35-55}$ TCR transgenic mice (C57BL/6-Tg(Tcra2D2,Tcrb2D2)1Kuch/J) and were originally purchased from Jackson Laboratory and maintained in-house. C3HeB/Fej (000658) were purchased from Jackson Laboratory as needed for experimentation. C57BL/6 (556) and SAS SD timed-pregnant rats were purchased from Charles River (SD400) for the purpose of postnatal OPC and glia cultures.

**Fig. 6** Caspase3/7 increases in OPCs that activate OT-1 CD8$^+$ T cells. **a** Fas surface expression on OPCs, either not stimulated (gray) or IFNγ stimulated (blue), as determined by flow cytometry for PDGFRα$^+$/CD11b$^-$ (FMO; white). Significance was determined by unpaired, two-tailed Student's t-test (P = 0.0046). Error bars represent standard deviation; 7 independent experiments. **b** Descriptive images of the cytotoxicity assay using NucLight stained OPCs and a Caspase 3/7 reporter. **c** Single cell controls of OT-1 CD8s (black circle) and OPCs (black square) cultured separately for a total of 60 hours with caspase 3/7 detection reagent and nuclear staining in OPCs. Representative images for 0 h, 24 h, and 48 h are shown and quantified from 0 to 60 h (**h**). **d** Time-lapse phase contrast and fluorescence imaging of OPC-CD8 co-cultures. At 2.0 h CD8s have an elongated/clustering morphology. At 20.0–28.0 h, CD8 cluster with the underlying caspase3/7 active OPC. **e** Time-lapse imaging of OPC-CD8 co-cultures; no antigen (gray triangle), ovalbumin protein (blue triangle), IFNγ stimulated OPC only (white circle). IFNγ+no antigen (gray circle) and IFNγ+ovalbumin protein (blue circle) [(IFNγ+no antigen vs. IFNγ+ovalbumin mean diff = −11410; 95% CI = −16239–(−)6581)] (**h**). **f** Total NucLight positive OPC quantification of the time course of the OPC-OT-1 co-culture. IFNγ stimulated OPCs that were pulsed with ovalbumin protein before the co-culture have a significant increase in number by 46 h. **g** At the time of CD8$^+$ T cell addition to OPCs stimulated with IFNγ+ovalbumin, pan caspase inhibitor (white circle;Q-VD-OPH; 10 μM) [(Vehicle vs. Q-VD-OPH mean diff = 21686; 95% CI = 15771–27601)], granzyme B inhibitor (green circle; 300 nM) [(Vehicle vs. GrB inhibitor mean diff = 16369; 95% CI = 10454–22284)] and FasL decoy receptor (pink circle; DcR3; 300 nM) [(Vehicle vs. DcR3 mean diff = 17751; 95% CI = 11836–23666)] were applied to the co-culture to determine the contribution of CD8 cytotoxicity pathways to OPC caspase activity. Significance of differences between conditions is shown in the legend for each figure. Significance for all quantified data was determined by area under the curve analysis followed by one-way ANOVA and Tukey's multiple comparisons test (P*≤0.05, **≤0.01, ***≤0.001, ****≤0.0001). All Error bars represent standard deviation from 3 to 5 experiments (unless otherwise specified)

*PDGFRα-CRE$^{ER}$* were provided by D. Bergles then backcrossed to the C57BL/6 background for 12 generations before being crossed with *Rosa26-YFP*. All animal protocols were approved and adhered to the guidelines of Johns Hopkins Institutional Animal Care and Use Committee.

**Recombinant cytokines and pharmacological inhibitors**. To determine the effects of cytokine on OPC differentiation and gene expression, cultures were supplemented with rat recombinant IFNγ (10 ng/mL; Peprotech) or rat recombinant IL-17 (50 ng/mL; R&D Systems). The antigen presentation studies mouse OPCs were used and cultured for 12 hrs (unless otherwise specified) with mouse recombinant IFNγ (10 ng/mL; Peprotech) in the presence of media supplemented with human recombinant PDGF-AA (20 ng/mL; R&D Systems). Pharmacological blockade of different aspects of the cytosolic MHC class I pathway was done using Chloroquine (100 μM; Sigma Aldrich), ONX-0914 (30 nM; ApexBio), and a Cathepsin S (10 nM; Calbiochem) inhibitor. Pharmacological blockade of CD8 mediated cytotoxicity pathways was examined using recombinant hDcR3, a decoy receptor for FasL (300 ng/mL; R&D Systems) and Granzyme B inhibitor, Z-AAD-CMK (300nM; Calbiochem). The broad-spectrum caspase inhibitor Q-VD-OPH (10 μM; Cayman Chemicals) was used to block caspase 3/7 reporter activity.

**Peptides**. Peptides or protein used for OPC presentation studies were MOG$_{37–50}$ (50 μg/mL; Johns Hopkins Synthesis and Sequencing Core), OVA$_{257–264}$ (50 μg/mL; Johns Hopkins Synthesis and Sequencing Core) or Ovalbumin protein (500 μg/mL; Sigma Aldrich).

MOG$_{1–125}$ (100 mg/mouse; MedImmune) was used for C3HEB/Fej immunizations. AT experiments used MOG$_{35–55}$ or MOG$_{97–114}$ (20 μg/mL; Johns Hopkins Synthesis and Sequencing Core) to select and reactivate isolated CD4 ex vivo.

**Primary cultures**. Cerebral cortices were dissected from P4-P6 rodent pups. To obtain cultures with high purity, OPCs were positively selected using A2B5 magnetic beads (Miltenyi) as previously described[36]. A2B5$^+$ cells were plated and expanded for 3–4 days with OPC media supplemented with recombinant human PDGF-AA until optimal density was reached and natural BMP antagonist Noggin to inhibit in vitro OPC to astrocyte differentiation and recombinant human PDGF-AA until optimal density was reached. If OPCs were being used in experimentation cells were maintained in PDGF-AA with the addition of cytokines or peptides. For differentiation assays, 96 h of culture combinations of T3 (10 nM; Sigma Aldrich) and recombinant cytokine were supplemented into OPC media.

For glia other than OPCs, cerebral cortices were dissected from P4 to P6 rat pups and were mechanically and enzymatically (2.5% trypsin) digested to obtain a single cell suspension before plating on poly-L lysine (PLL) coated T75 flasks. Mixed glia were cultured in glia medium (DMEM, high glucose with 10% heat-inactivated FBS (Hyclone) and 1% Penicillin/Streptomycin) for 8–9 days or until a multi-layer culture of astrocytes, OPCs and microglia was achieved. To remove microglia flasks were shaken at 180 rpm for 1 h. After shaking supernatant was removed and microglia were plated on PLL coated plates and cultured in microglia medium (2% heat-inactivated FBS). The medium was replenished and OPCs were shaken off for 8 h at 240 rpm. The supernatant containing OPCs was discarded and astrocytes were removed by trypsin then seeded onto PLL coated plates in glia medium (described above). In all, 24–48 h after plating microglia and astrocytes IFNγ stimulation followed by OVA$_{257-264}$ addition was begun (Identical experimental protocol as OPCs).

**CD8 isolation**. OVA peptide-specific CD8s were isolated from 8 to 12-week-old OT-1 transgenic mice. In brief, a single-cell suspension was prepared from spleen and lymph nodes. A CD8 negative selection was performed (Stemcell Technologies) and purified cells were stained with Cell Proliferation Dye eFluor450 (Fisher Scientific) according to the manufacturer's protocol. After flow analysis using Vβ5, CD8, CD62L, CD44 to test for purity and activation status, CD8s were then cultured with OPCs (see below).

**OPC-CD8 co-culture**. OPCs were prepared for co-culture (see above), then stimulated with IFNγ for 12 h (unless otherwise specified) to allow for transcription and translation of MHC class I antigen presentation pathway mediators. Next, OVA$_{257–264}$ or Ovalbumin was spiked into the medium and incubated for 8–12 h to permit for processing and presenting time. Cultures were washed so that the only source of antigen was already processed and presented by the OPCs. Isolated and stained OT-1 CD8s were plated with OPCs at a 3:1 CD8:OPC ratio in medium that was 50% CRPMI (RPMI 1640 (Invitrogen) +10% FBS +10 mM HEPES buffer (Quality Biological) +1 mM sodium pyruvate (Sigma-Aldrich) and MEM NEAA (Sigma-Aldrich) + βME + penicillin and streptomycin) and 50% OPC medium. Between 24 and 48 hours of culture, cells were collected and analyzed by flow cytometry (see below).

**Adoptive transfer – cuprizone studies**. *MOG$_{35–55}$ CD4 AT-CPZ into PDGFRα-Cre$^{ER}$ × Rosa26-YFP*: PDGFRα-Cre$^{ER}$ bred to the C57BL/6 background were crossed with Rosa26-YFP. At the age of 8–12 weeks crossed mice were transferred to a 0.2% CPZ diet for a total of 4 weeks. After 3 weeks of the CPZ diet, 4-hydroxytamoxifen (1 mg/mouse/day for 3 days) was injected to induce CRE recombination in PDGFRα expressing cells. MOG$_{35–55}$ specific CD4$^+$ T cells from 2D2 TCR transgenic mice were isolated, purified, polarized to TH17 subtype and expanded ex vivo as previously described[35]. The 2D2 CD4 purity was measured and ~8–10 million cells were injected, IP, into lineage tracing syngeneic recipient mice at the 4 or 6 week CPZ time point. Simultaneously the recipient mice were placed back on a normal chow diet (Supplementary Figs. 1 and 8). After 1–2 weeks, AT mice were killed and prepared for either IHC or flow cytometry analysis (below).

*MOG$_{97–114}$ AT-CPZ in C3HeB/Fej*: Donor C3HeB/Fej mice were immunized with 100 μg of whole rat MOG$_{1–125}$ protein in complete Freund's adjuvant (8 mg/mL heat-killed *Mycobacterium tuberculosis* + incomplete Freund's adjuvant). In all, 250 ng/mouse of Pertussis toxin was injected IP on the day of immunization and 2 days after immunization. After 9–10 days donor rMOG$_{1–125}$ immunized mice were killed, and spleen and lymph nodes were collected. A single cell suspension from these tissues was prepared. CD4$^+$ T cell negative selection was completed (StemCell Technologies) and cells were cultured ex vivo with irradiated APCs collected from the spleen of non-immunized C3HeB/Fej mice. In brief, 50 μg/mL MOG$_{97–114}$ peptide and 10 ng/mL of IL23 to expand MOG-specific T cell population and polarize these cells to the TH17 subtype was completed. After 3 days cells were collected and ficolled to remove dead cells. Approximately 10 million CD4+ T cells were adoptively transferred via an IP injection into recipient mice that were previously on a 0.2% CPZ diet for 6 weeks. Mice were monitored for EAE disease and once a score of 3 or greater was reached they were killed and prepared for flow cytometry analysis (below).

**Immunostaining**. Performed on cultured OPCs after IFNγ and OVA peptide incubation. Prior to fixation APC-α-H2Kb-OVA (1:100; Biolegend) was added to media after washing the cells. This antibody was incubated with live cells for 2 h

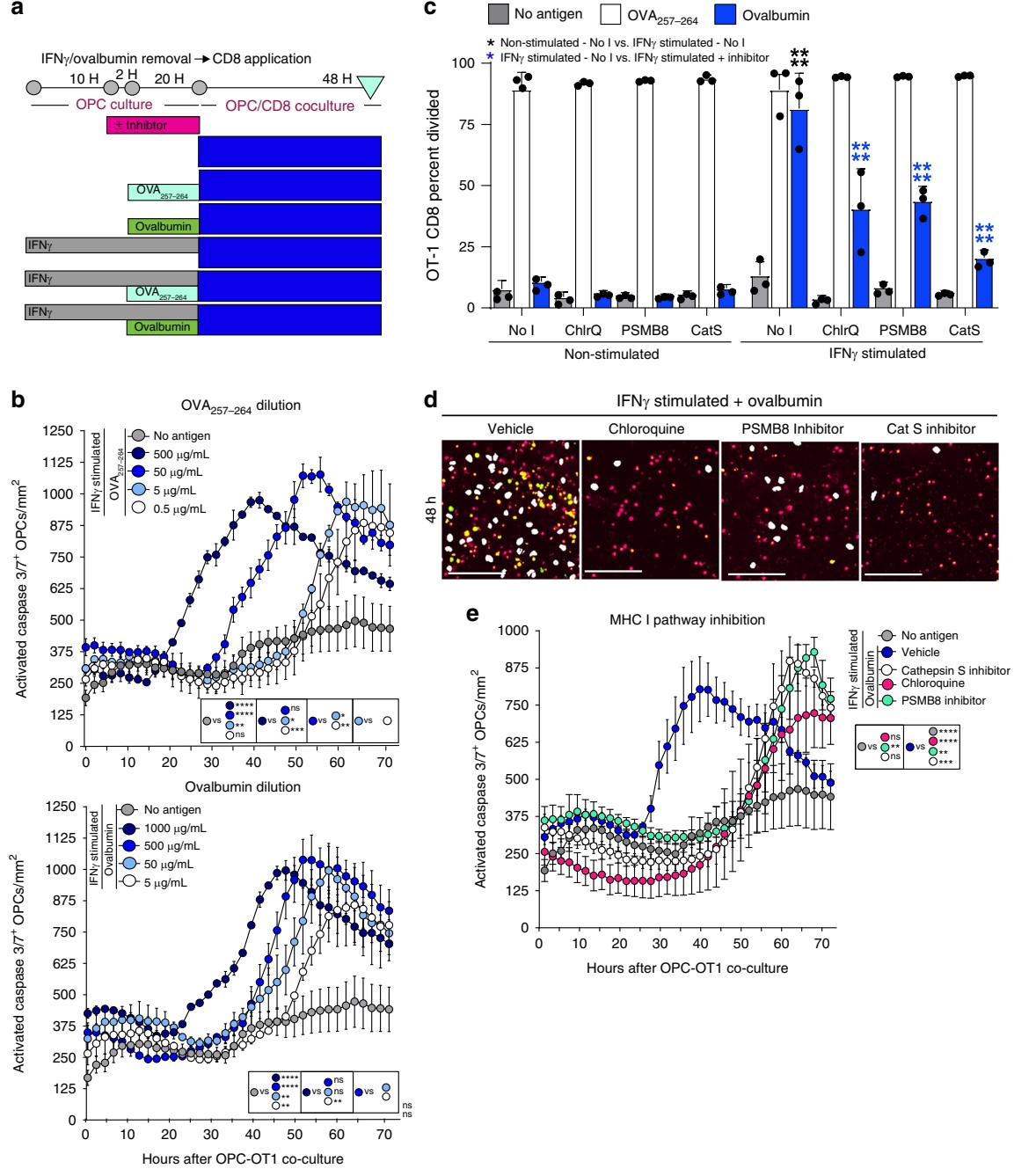

**Fig. 7** Antigen concentration and antigen pathway inhibitors modulate cytotoxicity. **a** Timeline of experimental methodology and treatment times.
**b** Quantification of time-lapse imaging of OVA$_{257-264}$ peptide (0.5–500 μg range) (top) [(500 μg/mL vs.5.0 μg/mL mean diff = 10320; 95% CI = 1902–18738)] and ovalbumin (5–1000 μg range) (bottom) [(1000 μg/mL vs.50 μg/mL mean diff = 12641; 95% CI = 4199–21083)] concentrations added to OPCs stimulated with IFNγ and antigen prior to CD8 addition. **c** Proliferation analysis of OPC-CD8 co-cultures treated with antigen cross-presentation pathway inhibitors. No inhibitor (No I) or each inhibitor were added to OPC culture for two hours prior to the addition of OVA$_{257-264}$ (white) or Ovalbumin (blue), and then left in wells during processing and presentation phase. Chloroquine was added at 100 μM. ONX-0914, PSMB8 inhibitor was added at 30 nM. Cathepsin S inhibitor was added at 10 nM. In all, 48 h after CD8 co-culture was initiated cell proliferation of the Vβ5, CD3, CD8a population of CD8s was analyzed for cell division and total percent divided. Significance for all quantified data was either assessed by two-way ANOVA analysis followed by a Tukey's multiple comparison analysis [(NoT vs. Chloroquine mean diff = 40.8; 95% CI = 27.1–54.4, NoT vs. ONX-0914 mean diff = 37.6; 95% CI = 24.0–51.3, NoT vs. Cathepsin S inhibitor mean diff = 60.9; 95% CI = 47.2–74.6)]. Error bars in all graphs represent standard deviation from three experiments. **d**, **e** Time-lapse images and quantification of from 0 to 60 h with the antigen presentation pathway inhibitors; Chloroquine (pink circle; 300 μM) [(mean diff = 18203; 95% CI = 11864–24542)], Cathepsin S inhibitor (white circle; 30 nM) [(mean diff = 9775; 95% CI = 3436–16114)], PSMB8 inhibitor (green circle; ONX-0914; 30 nM) [(mean diff = 13059; 95% CI = 6720–19398)]. Significance of differences between conditions is shown in the legend for each figure. Significance was determined by area under the curve analysis followed by one-way ANOVA and Tukey's multiple comparisons test. Error bars represent standard deviation from 3 to 5 experiments. (P*≤0.05, **≤ 0.01, ***≤0.001, ****≤0.0001)

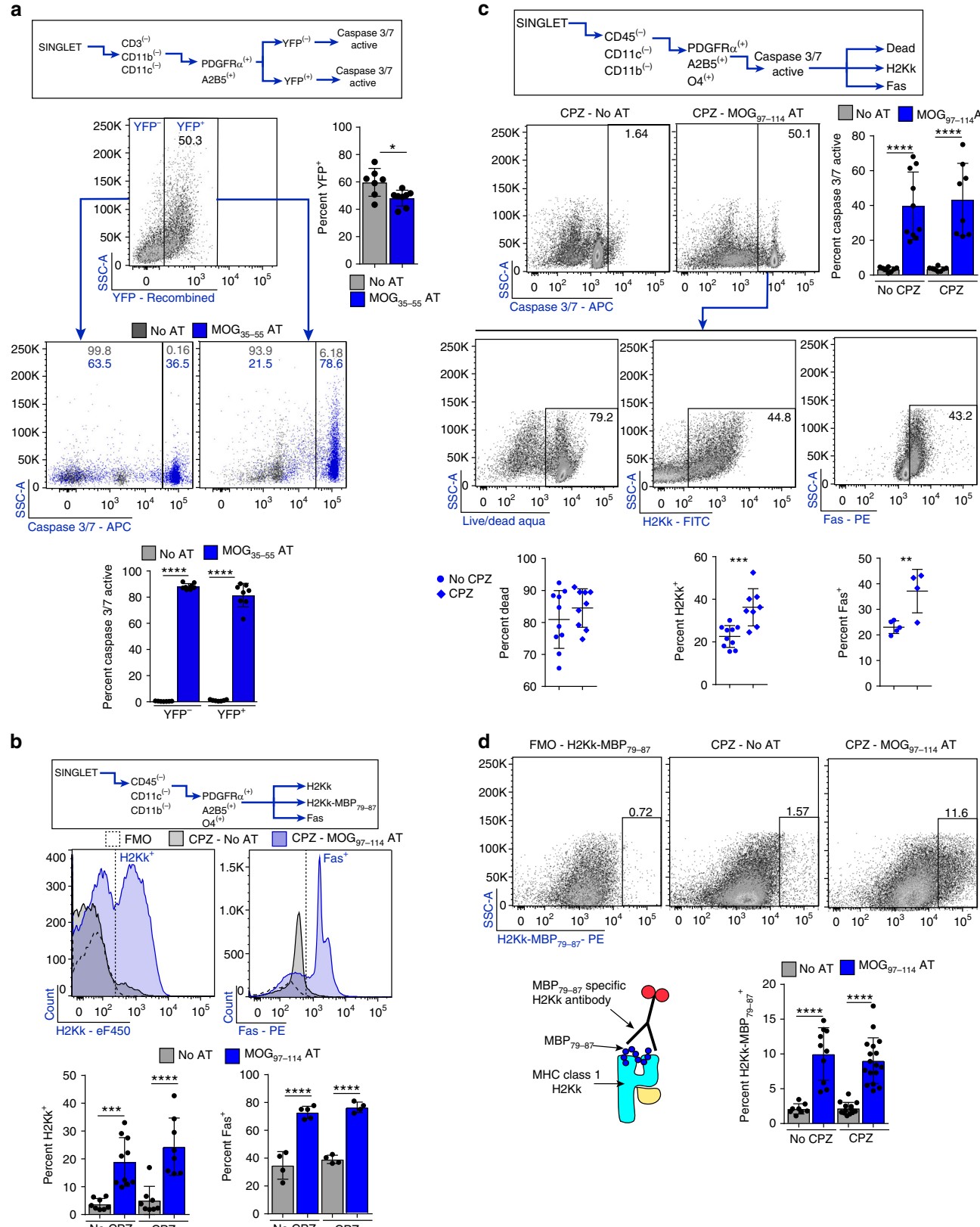

before fixation. After fixation α-rabbit-PDGFRα (1:1000; W. Stallcup; Burnham Institute) was used to verify culture purity and for colocalization purposes. DAPI counterstain was performed for 10 min then washed prior to slide mounting.

Mice were IP injected with sodium pentobarbital (250 μL of 5 mg/mL) and euthanized via cardiac perfusions with 30 mL PBS followed by 10 mL of 4% PFA. The whole brain was collected, post-fixed in 4% PFA for 12–18 hours, transferred

to 30% sucrose for 3 days, cryosectioned (30 μm) and kept in antifreeze, and stored in at −20 °C until staining was performed. Black Gold II (Millipore) was used to determine myelin content in the corpus callosum. Sections at ~−0.5 mm, −1.0 mm, and −2.0 mm posterior to bregma were stained with α-mouse- MBP SMI-99 (1:1000; eBioscience) and α-rabbit-CD3 (1:50; GeneTex) to determine myelin staining and lymphocyte infiltration, respectively. Lymphocyte characterization

**Fig. 8** OPCs express myelin peptide on MHC class I and are cytotoxic targets in vivo. C57BL/6 *PDGFRα-Cre$^{ER}$ × Rosa26-YFP* were kept on a CPZ diet for 4 weeks. After 3 weeks, 4-HT was injected to induce Cre recombination in PDGFRα expressing cells and MOG$_{35-55}$ reactive T cells were AT. Cells were isolated ex vivo and analyzed by flow cytometry. **a** The OPC population was distinguished based on YFP$^-$ and YFP$^+$ expression and analyzed for caspase 3/7 activity, quantification below; CPZ (gray) and CPZ+MOG$_{35-55}$ AT (blue). **b–d** Alternatively, C3HeB/FeJ donor mice were immunized with whole recombinant rat MOG$_{1-125}$ to induce EAE, and myelin-specific CD4 T cells were isolated ex vivo and reactivated with peptide. Syngeneic recipient mice were fed a CPZ diet for 6 weeks before AT. **b** H2Kk (left) [No CPZ vs. AT mean diff = −15.2, 95% CI = −24.7–(−)5.6, CPZ vs. CPZ+AT mean diff = −19.3, 95% CI = −29.4–(−)9.3], and Fas (right) (No CPZ vs. AT mean diff = −38.1, 95% CI = −49.6–(−)26.7, CPZ vs. CPZ+AT mean diff = −37.4, 95% CI = −49.5–(−)25.3) expression from the OPC population was determined by flow cytometry analysis with quantification below each flow histogram; no AT (gray; n = 8,8,4), MOG$_{97-114}$ AT (blue; n = 9,10,5), CPZ+no AT (gray; n= 8,12,4), and CPZ + MOG$_{97-114}$ AT (blue; n = 8,17,4). Statistical significance was determined by one-way ANOVA analysis followed by Tukey's multiple comparisons. Error bars represent the standard deviation. **c** Caspase 3/7 activity was determined by flow cytometry [No CPZ vs. AT mean diff = −36.4, 95% CI = −55.4–(−)17.5, CPZ vs. CPZ+AT mean diff = −39.9, 95% CI = −59.9–(−)20.0]. Further population analysis was done from the caspase 3/7 active population by Live/Dead staining (top), H2Kk expression (middle) (Diff between means = 13.7 ± 3.3, 95% CI = 6.7–20.7) and FAS expression (bottom) (Diff between means = 14.1 ± 3.9, 95% CI = 4.8–23.4). **d** H2Kk-MBP$_{79-87}$ staining from OPCs with FMO staining control and quantification below. At peak disease flow cytometry was completed on whole brain tissue to assess intracellular granzyme b staining in target cells (No CPZ vs. AT mean diff = −7.9, 95% CI = −11.3 to −4.5, CPZ vs. CPZ + AT mean diff = −6.9, 95% CI = −9.5 to −4.3)

and interaction was done using α-rat-CD3 (1:100; Genetex), α-rabbit-CD4 (1:50; Santa Cruz Biotechnology), and α-rabbit-CD8α (1:300; Abcam). Parallel sections were stained with chicken-α-YFP (1:1000; Abcam), rabbit-α-PDGFRα (1:600; W. Stallcup; Burnham Institute), and mouse-α-CC1 (1:50; Chemicon) to determine the fate of recombined OPCs. In addition, mouse-α-Olig2 (1:1000; Millipore) was used in combination with chicken-α-YFP to verify that PDGFRα driven CRE recombination within the corpus callosum was restricted to the oligodendrocyte lineage.

All brains were collected as part of the tissue procurement program approved by the Cleveland Clinic Institutional Review Board. All control and MS brain tissues were characterized for demyelination by immunostaining using 30 µm fixed tissue sections and proteolipid protein (PLP)[4]. Adjacent sections were used for double labeling using PLP/PSMB8 and SOX10/PSMB8. Primary antibodies used were rat anti-proteolipid protein (1:250, gift from Wendy Macklin, University of Colorado, Denver), goat anti-SOX-10 (1:100, R&D Systems, Minneapolis, MN), and rabbit anti-PSMB8 (1:1000, Thermo Fisher Scientific, Rockford, IL). Secondary antibodies were biotinylated donkey anti-rat IgG, donkey anti-goat IgG, and donkey anti-rabbit IgG (1:500, Vector Laboratories, Burlingame, CA), and Alexa 488 donkey anti-rat IgG, Alexa 488 donkey anti-goat IgG and Alexa 594 donkey anti-rabbit IgG (1:500, Invitrogen, Carlsbad, CA). Immunofluorescent-labeled tissues were analyzed using a Leica DM5500 upright microscope (Leica Microsystems, Exton, PA). PLP-PSMB8 and PSMB8-Sox10 images were collected from corresponding sections and manually aligned. Resultant images were analyzed using Fiji version (http://fiji.sc) of the free image processing software ImageJ (NIH, http://rsbweb.nih.gov/ij) to evaluate co-localized PSMB8 and SOX10 positive cells and presented as cells/mm2.

**Proximity analysis**. To determine the interaction between YFP$^+$ OPCs and T-lymphocytes AT-CPZ mice coronal sections were stained as described above. Confocal microscopy was completed using a Zeiss LSM 800 microscope (Multi-photon Imaging Core). Post-acquisition analysis to determine the percent of OPCs that were interacting with either CD3$^+$/CD4$^+$ or CD3$^+$/CD8$^+$ was executed using Imaris software (Bitplane). In brief, T cells were identified as spots and the YFP$^+$ staining was used to create the surface mask. The spots close to surface analysis extension was used to count the number of T cell 3.0 µm or less from the center of the spot (T cell) to the surface of YFP surface mask. The number of OPCs were counted manually from each image analyzed.

**Flow cytometry**. OPCs or CD8s cultured in vitro were removed from their culture vessel by repeated pipetting. For staining panels that interrogated cytokine and granular protein production cell stimulation was done for 8 h using Cell Stimulation Cocktail with protein transport inhibitors (1:250; eBioscience). Live/Dead staining using LIVE/DEAD Fixable Aqua Dead Cell Stain Kit (Thermofisher) was performed for 20 min. Surface staining using the following CD8 T cell markers; APC-CD8a (1:100; eBioscience), PE-α-Vβ5 (1:100; Biolegend), and APC eFluor 780-α-CD3 (1:100; eBioscience) was completed for 30 minutes. In addition, surface markers for T cell activation were used; APC eFluor 780-α-CD44 (1:100; eBioscience), APC-α-CD62L (1:100; Biolegend), PerCP Cy5.5-α-CD25 (1:100; Biolegend), and FITC-α-CD69 (1:100; Biolegend). Cell permeabilization and fixation was done using IC Fixation Buffer (ThermoFisher) for 20 min. Intracellular staining was performed using FITC, PerCP Cy5.5-α-IFNγ (1:100; Biolegend), PE Cy7-α-TNFα (1:100; Biolegend), APC-α-IL-17 (1:100; Biolegend), FITC-α-GM-CSF (1:100; eBioscience), PE-Cy7-α-Granzyme B (1:50/1:100; Biolegend), and PE-α-perforin (1:100; Biolegend). Staining panels that did not require intracellular staining were immediately stained with a LIVE/DEAD Fixable Aqua Dead Cell Stain Kit (ThermoFisher) after removal from the culture vessel. Surface staining for

OPCs was completed using PE Cy7-α-PDGFRa (Miltenyi; 1:25), APC-α-A2B5 (Miltenyi; 1:25), PerCP Cy5.5-α-CD11b (Biolegend; 1:100), BV-421-α-CD11c (Biolegend; (1:100), PE-α-Fas (eBioscience; 1:100), and FITC-α-H2Kb (eBioscience 1:100). Simultaneously staining using Caspase 3/7 Green or Red reagent (Essen Biosciences) was completed for 1 h. Data represented in diagrams and analysis is negative for CD11b and CD11c and positive for PDGFRα and A2B5. Stained cells were run on an 8 color MACSQuant Analyzer (Miltenyi).

AT-CPZ mice were IP injected with sodium pentobarbital (250 µL of 5 mg/mL), after which a cardiac perfusion with 30 mL of PBS was completed. The whole brain was isolated and finely chopped to prepare for efficient enzymatic digestion of tissue using Neural Tissue Dissociation Kit with papain (Miltenyi) according to the manufacturer's protocol. The remaining tissue was further processed through a 100-µm pore plastic filter to collect a single cell suspension from each brain sample. The myelin component and debris were removed by a 35% percoll gradient centrifugation for 20 minutes, no brake at 650 × g. The cell pellet was resuspended and used for flow cytometry staining. Live/Dead Aqua staining (Thermo Fisher) was performed for 20 min followed by a mouse CD16/CD32 Fc block (1:50; Biolegend) for 10 min. To analyze T cell infiltration APC-eFluor 780-α-CD3 (1:100; Biolegend), eFluor 450-α-CD4 (1:100; Biolegend), APC-α-CD8 (1:100; Biolegend) were used. Dendritic cell, macrophages and microglia populations were analyzed based on the expression of APC-eFluor 780-α-CD45 (1:100; eBioscience), BV-421, PE-α-CD11c (1:100; Biolegend), and PerCP Cy5.5-α-CD11b (1:100; Biolegend). The OPC population was identified using PE Cy7-α-PDGFRα (1:25; Miltenyi) in combination with APC-α-A2B5 (1:25; Miltenyi) or PE-α-O4 (1:25; Miltenyi) in combination with APC-α-A2B5. General MHC class I presentation staining was done for mice on the C57BL/6 and C3HeB/Fej backgrounds using PE,APC-α-H2Kb (1:100; eBioscience) and FITC,eFluor 450-α-H2Kk (1:100; Thermo Fisher), respectively. MBP$_{79-87}$ specific H2Kk staining was performed using PE, FITC-αH2Kk-MBP$_{79-87}$ (J. Goverman/Rockland). Finally, to identify cells undergoing cell death and Fas expression interrogation Caspase 3/7 Green or Red reagent (1:1000; Essen Bioscience) and PE-α-Fas (1:100; BD Biosciences) were utilized. Stained samples were analyzed on an 8 color MACSQuant Analyzer (Miltenyi).

**Gene array analysis**. Affymetrix microarrays were completed on control and IFNγ treated OPC cultures at baseline (PDGF D0), 8 h, 24 h, 48 h, and 96 h after IFNγ stimulation was initiated. Gene set enrichment analysis (GSEA)[69] was done by comparing T3 treated samples to T3 + IFNγ treated samples to determine pathway enrichment for genes associated with IFNγ signaling. Data from GSEA was and used in the targeted heat map (Fig. 2e).

**Live cell imaging**. OPCs were stained with an eFluor 633 dye prior to the addition of CD8s to ensure subsequent quantification was restricted to the OPC population. Unstained OT-1 CD8s were isolated and added to the stimulated and stained OPCs. Cell membrane permeable, caspase 3/7 Green reagent (1:1000; Essen Biosciences) was added to determine caspase activity. The caspase 3/7 reported was designed to report on caspase 3/7 activity by targeted cleavage of the DEVD peptide sequence and subsequent nuclear translocation of the linked DNA binding dye (Essen Biosciences). Images were captured every 2–3 h using IncuCyte S3 Live Cell Analysis System (Sartorius). Cultures were analyzed for caspase 3/7 stain coloca-lization with NucLight Rapid Red (Essen Biosciences) labeled OPCs and the applied mask was counted as particles per image with 9 images taken. A total of three experiments were completed. MHC class I inhibitors were applied two hours before antigen was given to IFNγ stimulated OPCs and kept in culture for the antigen loading portion of the experiment until CD8s were applied and antigen, IFNγ and inhibitors were washed away from the culture. Fas/FasL and Granzyme B

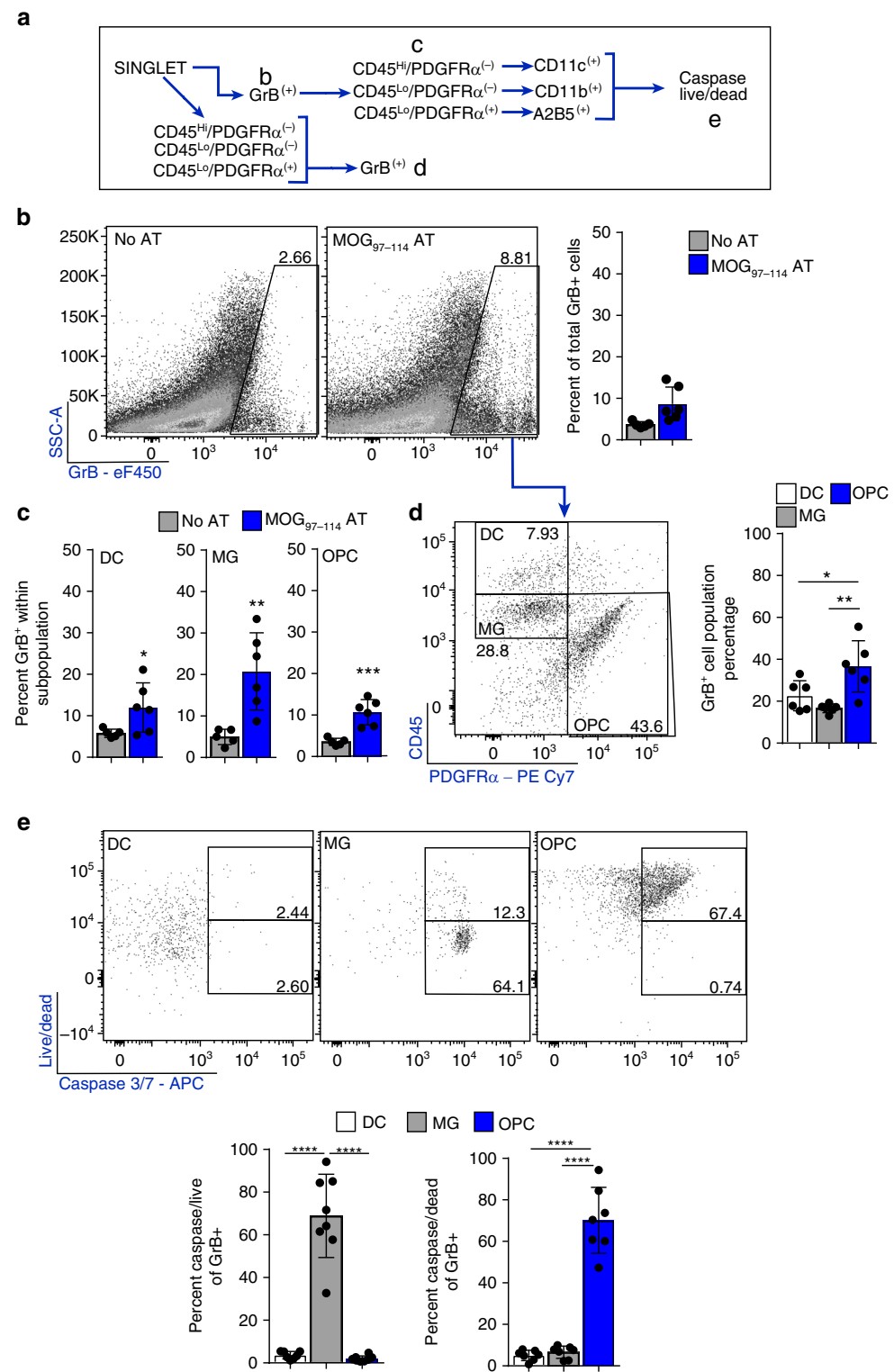

inhibitors were applied to the culture at the time of OPC-CD8 co-culture initiation and were kept in culture for the remainder of the experiment.

**Quantitative PCR**. RNA was extracted from cultured OPCs using RNeasy Plus Mini Kit (Qiagen). First-strand cDNA was synthesized using iScript cDNA Synthesis Kit (Bio-Rad). Quantitative PCR was completed using SensiMix (Bioline) a SYBR based reagent and a CFX384 Touch Real-Time PCR Detection System (Bio-Rad). All target genes were normalized to *hprt1* reference gene and delta-delta CT analysis was performed.

**Statistical analysis**. Unpaired two-tailed Student's *t*-test, Pearson Correlation, one-way ANOVA with Tukey's multiple comparisons post hoc test and two-way ANOVA with Tukey's multiple comparisons post hoc test were performed. Specific tests are noted in figure legends with significance level annotation. All error bars represent the standard deviation of the mean.

**Approvals**. The research reported in this study complied with all relevant ethical regulations for animal testing and research and was approved by the Johns Hopkins Institutional Animal Care and Use Committee. All MS brain tissues used for

**Fig. 9** OPCs are targets for granzyme B mediated cytotoxicity in vivo. C3HeB/FeJ donor mice were immunized with whole recombinant rat MOG$_{1-125}$ to induce EAE, and myelin-specific CD4 T cells were isolated ex vivo and reactivated with peptide. Syngeneic recipient mice received MOG$_{97-114}$ T cell AT then killed at peak disease. **a** Cells gating strategy for flow cytometry analysis. **b** Total granzyme b staining from the initial cell and singlet gates. Percent granzyme B positive staining is compared between No AT (gray; $n = 5$) and MOG$_{97-114}$ AT (blue; $n = 6$) (Diff between means = 5.0 ± 1.9, 95% CI = 0.8–9.2). **c** Percent granzyme B staining within designated target cell populations were compared between No AT (gray; $n = 5$) and MOG$_{97-114}$ AT (blue; $n = 6$) (DC diff between mean = 6.2 ± 2.7, 95% CI = 0.1-12.4, Microglia diff between means = 15.8 ± 4.3, 95% CI = 6.2-25.4, OPC diff between means = 7.1 ± 1.4, 95% CI = 3.9-10.4). **d** Total granzyme B positive cell population, within the AT group, was identified using PDGFRα and CD45 levels and quantified below (DC vs. microglia mean diff = −25.8, 95% CI = −41.2-(−)10.5, DC vs. OPC mean diff = −18.6, 95% CI = −33.9-(−)3.2). **e** DC, MG, and OPCs populations from the total granzyme B positive population was assessed by caspase 3/7 reporter and live/dead staining to determine viability of each population and quantified for caspase$^+$/live and caspase$^+$/dead populations as shown by bottom and top box gates (respectively) in flow plots. Quantification of data is shown below (caspase$^+$/dead; microglia vs. OPC mean diff = −65.2, 95% CI = −75.9-(−)54.5, DC vs. OPC mean diff = −66.0, 95% CI = −76.7-(−)55.3). Significance for all quantified data was either assessed by one-way ANOVA analysis followed by Tukey's multiple comparison analysis or unpaired, two-tailed $t$-test ($P^* \leq 0.05$, $** \leq 0.01$, $*** \leq 0.001$, $**** \leq 0.0001$). Error bars in all graphs represent standard deviation

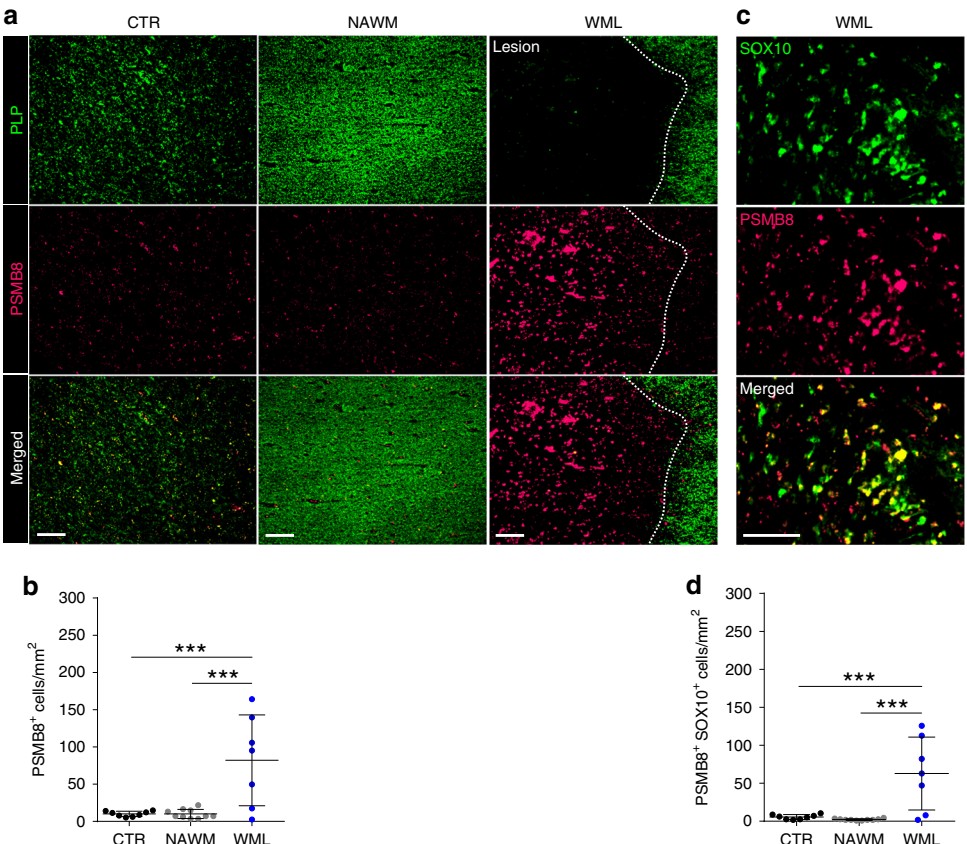

**Fig. 10** MS lesions have high PSMB8 expression in oligodendrocytes lineage cells. **a** Postmortem tissue from healthy controls or MS patients' normal appearing white matter (NAWM) and white matter lesions (WML) was analyzed for the immunoproteasome specific subunit, PSMB8 and proteolipid protein (PLP) (scale bar 200 μm). **b** PSMB8 staining colocalization with SOX10 staining. **c**, **d** Quantification of imaging. A total of eight healthy controls, ten NAWM and seven WML sections were counted. Significance was determined by one-way ANOVA analysis followed by Tukey's multiple comparison analysis ($P^* \leq 0.05$, $** \leq 0.01$, $*** \leq 0.001$, $**** \leq 0.0001$). Error bars in all graphs represent standard deviation

the staining were de-identified postmortem brain tissue and therefore did not constitute human subjects research and were exempted from IRB.

### Data availability

The microarray data are available at GEO Submission (GSE133508) [NCBI tracking system #20113763] https://www.ncbi.nlm.nih.gov/geo/query/acc.cgi?acc=GSE133508

The datasets generated during and/or analyzed during the current study are available from the corresponding author on reasonable request.

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

## Acknowledgements

This research was supported by US National Institutes of Health grant NS-R37041435 (PAC), a NMSS Collaborative Center Award (PAC), The Race to Erase MS (PAC), and MedImmune (PAC); and the Miriam and Sheldon Adelson Medical Research Foundation (DB). We would like to thank Dr. Brian Popko (University of Chicago) for kindly providing us with the TRE/IFNγ and GFAP/tTA transgenic mouse lines and Dr. William Stallcup (Burnham Institute) for kindly providing the α-rabbit-PDGFRα antibody. We would like to thank the Solomon H. Snyder Multiphoton Imaging Core supported by the US National Institute of Health (NS050274) for use of confocal microscopes and Imaris software (Bitplane).

## Author contributions

L.K. and P.A.C. designed experiments. L.K. executed and analyzed the majority of experiments described in figures and text (unless otherwise stated). J.J. and M.D.S executed initial experiments with AT-Cup lineage tracing in Fig. 1 and Supplemental Figs. 1 and 2. J.G.C. completed quantitative PCR validation of microarray findings in Fig. 2. K.A. M., H.S., L.H., and M.A. managed mouse husbandry, genotyping, and phenotyping. J.W. analyzed microarray and performed GSEA. R.D. provided data and analysis for MS patient tissue staining in Fig. 10. D.B., J.G., T.D., J.K., and M.D.S provided critical and continual feedback regarding experimentation and data analysis. J.G., T.D., and J.K., created and provided important reagents used in experiments. L.K. and P.A.C. wrote the manuscript.

## Additional information

**Competing interests:** The authors declare no competing interests.

