## [Peer Review File · Nature Communications]

Reviewers' comments:

Reviewer #1, expert on myelination (Remarks to the Author):

In this manuscript, Kirby et al. describe the results of a very comprehensive set of cell-based and in vivo studies that demonstrate the ability of oligodendrocyte precursor cells to cross present antigen in a cell-type and defined context specific manner. The authors demonstrate, using both cell-based assays and diverse transgenic in vivo mouse models, that in response to defined cytokine(s) (i.e., IFN γ , but not IL-17), OPCs upregulate MHC class-1 (and -2) and cross present exogenous antigens (in a cytosolic processing pathway specific manner) to cytotoxic CD8 T-cells. This process is demonstrated to cause subsequent caspase 3/7 activation and death in OPCs.

Specifically, to determine how CD4+ T-cell effector function specifically impacts OPC differentiation, the authors performed expression profiling (and protein analysis) on primary mouse cortical postnatal OPCs, which resulted in the finding that IFN γ induces the expression of antigen processing and cross-presentation pathways. This phenomena is then confirmed in vivo, using a transgenic mouse model of CNS restricted IFN γ expression, in which it is demonstrated that IFN γ induces MHC class-1 and -2 in OPCs. Using TAP-/- vs WT mouse-derived OPCs, the authors demonstrate that OPCs process and present exogenous antigens on H2Kb (MHC class-1) via the cytosolic pathway. Using PDGFR α -CRE x Rosa26-YFP cuprizone fed mice, combined with adoptive transfer of MOG-specific T-Cells, it is demonstrated that these processes occur under disease relevant conditions in vivo. In response to T cell infiltration, it is demonstrated that MHC class I is expressed in OPCs and loaded with myelin peptides, which are then targeted for CD8-mediated destruction. Finally, by performing IHC on human tissue samples from control and MS patients, it is found that this phenomenon likely occurs in MS human patients. Immunoproteasome-specific subunit PSMB8 is demonstrated to be differentially prevalent in white matter lesions, as compared to normal white matter or control patient tissues.

Very recently, Castelo-Branco et al. have reported the identification of disease-specific oligodendrocyte lineages in the context of the EAE disease mouse model of MS (Nature Medicine, published online Nov. 12 2018). In this report, MHC class-1 and -2 related genes are found to be upregulated in a subset of OPCs in the context of EAE disease. The Castelo-Branco manuscript is more of a qualitative assessment of lineages and does not provide detailed molecular levels insights into the putative role of OPCs in the context of antigen cross presentation in a cytokine-specific manner. Further, the Castelo-Branco manuscript does not provide bridging human patient-based data. As such, it is the opinion of the reviewer, that this manuscript is complimentary and does not detract from the novelty of the manuscript by Kirby et al..

Overall, the findings of this manuscript have broad implications for the development of more effective treatment strategies for immune-based demyelinating diseases (e.g., multiple sclerosis). Specifically, the results provide defined and specific targets and strategies that will enable the development of agents and strategies that directly target the underlying cause of a perpetuated pathological T-cell-mediated immunological responses within the CNS. The extensive amount of data that is presented strongly supports the overall conclusions of the manuscript. Further, the findings have broad implications that will be of interest to a diverse readership. As such, following minor revision, this manuscript should be published in Nature Communications.

Minor points to address:

1. In Fig. 1G it is hard to determine the level of overlap between YFP+ oligodendroglial cells and either CD4+ or CD8+ cells, based on the presented data. Is it possible to provide a quantitative assessment of these phenomena for comparative and relative purposes?
2. It is unclear why supplementary Fig. 5 is not part of main Figure 2?

3. It appears that the text describing Figure 3C does not match the data presented in the Figure (i.e., 97% MHC class I hi vs 4% and 2% in pOL vs mOL). In the bar graph pOL is ~50% and on the histogram it is 67.8%. This must be corrected or clarified. This is a potential significant issue, as a major finding of this paper is the apparent differential ability of OPC vs mOL to cross present antigen.

4. In figure 6, it is presumed, based on image analysis, that Casp3/7 signal is not derived from OT1 CD8 cells. However, the level of casp3/7 in CD8s does not appear to have been quantified in Fig. 6B. What is the impact of IFN γ (+/- antigen) on CD8s? As the stained OPC red channel is completely saturated (by contrasting), and cell densities are high, it is hard to determine defined OPCs that are Casp3/7 positive, as presented. Based on the overlap mask, however, there appears to be a significant difference between Ova and no antigen following IFN γ stimulated (key takeaway). However, increased clarity around this result would benefit the manuscript.

5. Please clarify why there is a significant induction of Casp3/7 in OPCs in the presence of ovalbumin in the absence of IFN γ stimulation at 48 hours, in Fig. 6C.

6. X-axis title is missing from Figure 6D.

Reviewer #2, expert on EAE (Remarks to the Author):

The focus of this manuscript is regarding the role of oligodendrocyte precursor cells (OPCs) in immune/autoimmune processes in the CNS. The authors explore phenotypic and functional changes of OPCs during autoimmune neuroinflammation. The main overall finding is that OPCs exposed to inflammatory milieu can develop characteristics similar to professional antigen presenting cells (APCs) and perpetuate or amplify inflammatory processes in affected CNS areas. The authors identify IFN-g as an important inducer of APC-like phenotype in OPCs. IFN-g induces immunoproteasome and MHC class I expression in OPCs, which enables antigen cross presentation by OPCs to CD8 T cells, leading to activation and proliferation of CD8 T cells and death of OPCs by caspase 3/7 activation. Exposure of OPCs to IFN-g (and IL-17) also precludes differentiation of OPCs into mature myelinating oligodendrocytes. Hence, IFN-g contributes to pathological processes in CNS autoimmunity by amplifying inflammation via OPCs, which leads to their death, and by blocking the development of OPCs into myelinating cells.

The manuscript is well written clearly presenting findings and competently discussing them. However, Methods contain descriptions of experiments with rats. It appears that these experiments are not present in the current version of the manuscript.

Finding that OPCs can be readily driven toward "immune phenotype" is novel and highly relevant for understanding pathogenic mechanisms of CNS autoimmunity. These new insights have a potential to clarify mechanisms that underlie chronicity of autoimmune neuroinflammation and failure of spontaneous remyelination.

The principal deficiency of the manuscript is the lack of experiments that test the role of IFN-g signaling in OPCs in the context of autoimmune neuroinflammation (EAE) in vivo. This would perhaps be best achieved with the use of conditional knockout mice lacking IFN-g receptor in OPCs. Such experimental design should be feasible given that conditional IFN-g receptor knockout mice are commercially available. The relevance of IFN-g signaling in OPCs in autoimmune neuroinflammation without in vivo testing remains unclear given that IFN-g has net anti-inflammatory role.

Reviewer #3, expert on antigen presentation (Remarks to the Author):

In this ms, Kirby et al. establish multiple

Effector CD4+ T cells inhibit Oligodendrocyte precursor cells (OPCs) differentiation and myelin production

IFN γ expression in the CNS is sufficient to inhibit Oligodendrocyte precursor cells (OPCs) differentiation and myelin production

IFN γ induce a transcriptional response in OPC including the up regulation of immunoproteasome and MHC I expression

T cell exposed or IFN-responsive OPC acquire the ability to cross present antigens by MHC I.

Immunoproteasome expression in oligodendrocytes is a biomarker associated to demyelinated white matter.

From this the authors propose that OPC may be coopted to amplify and perpetuate the local auto-immune reaction in MS, upon IFN γ -dependent activation and possibly through cross-presentation by MHC I to auto-reactive CD8+ T cells .

Strength:

The study is nicely performed and carefully presented. The results are clear. In particular the data establishing the activation of the MHC I cross-presentation by IFN γ stimulation OPCs are convincing. In sum, this reports a new finding which may have an important significance in MS and brain infections. This highlights the possible role of OPC in local antigen cross presentation by MHC I.

Weakness:

The immunopathological relevance of cross-presentation by MHC I for immunopathology is not assessed. Indeed, Th17-dependent immunopathology is largely dependent on the GM-CSF released by T cells which in turn activate monocytes to differentiate into inflammatory macrophages mediating tissue damage (Codarri L, Gyölvésszi G, Tosevski V, Hesske L, Fontana A, Magnenat L, Suter T, Becher B. *Nat Immunol.* 2011 Jun; 12(6):560-7; Spath S, Komuczki J, Hermann M, Pelczar P, Mair F, Schreiner B, Becher B. *Immunity.* 2017 Feb 21; 46(2):245-260., Croxford AL, Lanzinger M, Hartmann FJ, Schreiner B, Mair F, Pelczar P, Clausen BE, Jung S, Greter M, Becher B. *Immunity.* 2015 Sep 15; 43(3):502-14. Codarri L, Gyölvésszi G, Tosevski V, Hesske L, Fontana A, Magnenat L, Suter T, Becher B. *Nat Immunol.* 2011 Jun; 12(6):560-7).

The role of CD8+ T cells-dependent elimination of PCs via cross presentation is not incompatible with this view but would deserve some functional validation. For example, evidencing that CD8+ T cells impact on OPCs upon adoptive transfer (MOG97_114) by depleting CD8+ T cells with antibodies. It is anticipated that depletion of CD8+ T cell should increase the population of Fas+K κ + OPCs and reduce the onset of caspase3/7 activation.

The immunopathological contribution of IFN γ signalling in OPCs is not tested in vivo. The study would be strengthened by providing a genetic evidence in vivo. This could be achieved by showing that conditional inactivation of IFN γ R or STAT1 in OPCs confers a significant protection from demyelination. Or minimally to evidence that IFN γ antibody blockade prevents the OPC activation and their elimination by CD8+ T cells in the context of adoptive transfer of Th17 CD4+ T cells.

In summary, this is an important study which i) provides an explanatory framework for the predominance of CD8+ T cells in MS lesions; ii) identifies new target for the therapeutic purpose of preventing demyelination

However the study falls short in providing a compelling in vivo evidence for the relevance of the findings obtained by in vitro analysis.

Specific comments:

Figure 5 : the authors show that inhibitors of cross-presentation do not inhibit OT1 activation by OPC: this should be tested by titrating the SIINFEKL peptide across a broad range of concentrations.

Figure 6 : The authors show that Fas expression is induced by IFN γ on OPCs but its contribution to cell death of OPC is not tested.

Figure 6 : The number of remaining live ODC should be monitored to directly assess cell death induced by cross presentation

Figure 6 : do inhibitors of cross presentation (clhrq, ONX, CatS, cf fig. 5) inhibit caspase3/7 activation and cell death induced by OTI?

Figure 7d : the experiments should be better controlled using an isotope-matched control for the staining of Kk79-87 in all 4 conditions (+/-AT, +/-CPZ).

May 6, 2019

Dear Reviewers,

Thank you for your thoughtful comments. We appreciate the high level of enthusiasm expressed by all of the reviewers and have performed several additional experiments to address your concerns. Please see our point-by-point responses below. Overall, we believe that the manuscript has been significantly improved through the review process and look forward to your response.

Response to Reviewers: Provided in Bold

Reviewer #1, expert on myelination (Remarks to the Author):

Very recently, Castelo-Branco et al. have reported the identification of disease-specific oligodendrocyte lineages in the context of the EAE disease mouse model of MS (Nature Medicine, published online Nov. 12 2018). In this report, MHC class-1 and -2 related genes are found to be upregulated in a subset of OPCs in the context of EAE disease. The Castelo-Branco manuscript is more of a qualitative assessment of lineages and does not provide detailed molecular levels insights into the putative role of OPCs in the context of antigen cross presentation in a cytokine-specific manner. Further, the Castelo-Branco manuscript does not provide bridging human patient-based data. As such, it is the opinion of the reviewer, that this manuscript is complimentary and does not detract from the novelty of the manuscript by Kirby et al..

Overall, the findings of this manuscript have broad implications for the development of more effective treatment strategies for immune-based demyelinating diseases (e.g., multiple sclerosis). Specifically, the results provide defined and specific targets and strategies that will enable the development of agents and strategies that directly target the underlying cause of a perpetuated pathological T-cell-mediated immunological responses within the CNS. The extensive amount of data that is presented strongly supports the overall conclusions of the manuscript. Further, the findings have broad implications that will be of interest to a diverse readership. As such, following minor revision, this manuscript should be published in Nature Communications.

We greatly appreciate the reviewer's recognition of the value of our manuscript to the literature.

Minor points to address:

1. In Fig. 1G it is hard to determine the level of overlap between YFP+ oligodendroglial cells and either CD4+ or CD8+ cells, based on the presented data. Is it possible to provide a quantitative assessment of these phenomena for comparative and relative purposes?

We now provide a quantitative assessment using Imaris Bitplane software to quantify the proximity and overlap between YFP+ OL lineage cells and CD4+ or CD8+ Cells at 1 week and 2 weeks post adoptive transfer (Figure 1h).

2. *It is unclear why supplementary Fig. 5 is not part of main Figure 2?*

This is an excellent point and we have now fit supplementary Figure 5 into Figure 2f.

3. *It appears that the text describing Figure 3C does not match the data presented in the Figure (i.e., 97% MHC class I hi vs 4% and 2% in pOL vs mOL). In the bar graph pOL is ~50% and on the histogram it is 67.8%. This must be corrected or clarified. This is a potential significant issue, as a major finding of this paper is the apparent differential ability of OPC vs mOL to cross present antigen.*

We apologize for this oversight and have corrected the text, which was inaccurate.

4. *In figure 6, it is presumed, based on image analysis, that Casp3/7 signal is not derived from OT1 CD8 cells. However, the level of casp3/7 in CD8s does not appear to have been quantified in Fig. 6B. What is the impact of IFNg (+/- antigen) on CD8s? As the stained OPC red channel is completely saturated (by contrasting), and cell densities are high, it is hard to determine defined OPCs that are Casp3/7 positive, as presented. Based on the overlap mask, however, there appears to be a significant difference between Ova and no antigen following IFNg stimulated (key takeaway). However, increased clarity around this result would benefit the manuscript.*

Thank you for raising this concern. We have addressed this issue using several strategies to provide more confidence in our caspase 3/7 reporter readout in OPCs.

- 1. Rather than using a cytoplasmic stain to track the OPCs we have optimized and switched to using NuLight Rapid Red reagent that is applied to the OPCs before addition of CD8s such that no CD8s are stained. Further, we saw no evidence of CD8s picking up unwashed stain during the coculture. This stain is constrained to the nucleus of the labeled OPCs where activated caspase 3/7 fluorescence appears when the cell is undergoing apoptosis.**
- 2. We did observe a number of small particles that were double positive. Based on size, these particles are cell debris. While this was a confounding factor in our previous analysis we have incorporated a size restriction so that small particles would not be counted.**
- 3. In Figure 6e, using IncuCyte live cell analysis imaging, we have provided an example of the time course and apoptosis of an OPC using fluorescence and phase contrast imaging.**
- 4. We have also quantified the total NuLight positive OPCs and find that the caspase 3/7 activity corresponds to a significant reduction in this population consistent with OPC death (Fig. 6d).**
- 5. Finally, we show significant blockade of apoptosis using DcR3 (FasL inhibitor) and granzyme B inhibitor as compared to the IFNg stimulated OPC plus Ovalbumin but not in the no antigen control. These results further support the relevance of these specific cell death pathways in OPCs (Fig 6g,h).**

5. Please clarify why there is a significant induction of Casp3/7 in OPCs in the presence of ovalbumin in the absence of IFN γ stimulation at 48 hours, in Fig. 6C.

Thank you for this comment. We agree the old version was a bit confusing as the cellular origin of the activated Caspase 3/7 signal was ambiguous. We now show the proportion of OPCs with activated Caspase 3/7. There is a significant increase in Caspase 3/7+ OPCs with IFN γ and antigen as compared to control conditions (Fig 6h). We have included more statistical analyses between control groups and show that there is not statistical significance between non stimulated + antigen and non-stimulated control.

6. X-axis title is missing from Figure 6D.

An X-axis title has now been added.

Reviewer #2, expert on EAE (Remarks to the Author):

The manuscript is well written clearly presenting findings and competently discussing them. However, Methods contain descriptions of experiments with rats. It appears that these experiments are not present in the current version of the manuscript.

We apologize for the confusion, the gene expression array and qPCR data in figure 2 was done in rats since they may be more similar to humans. All other experiments are in mice.

Finding that OPCs can be readily driven toward “immune phenotype” is novel and highly relevant for understanding pathogenic mechanisms of CNS autoimmunity. These new insights have a potential to clarify mechanisms that underlie chronicity of autoimmune neuroinflammation and failure of spontaneous remyelination.

Thank you

The principal deficiency of the manuscript is the lack of experiments that test the role of IFN-g signaling in OPCs in the context of autoimmune neuroinflammation (EAE) in vivo. This would perhaps be best achieved with the use of conditional knockout mice lacking IFN-g receptor in OPCs. Such experimental design should be feasible given that conditional IFN-g receptor knockout mice are commercially available. The relevance of IFN-g signaling in OPCs in autoimmune neuroinflammation without in vivo testing remains unclear given that IFN-g has net anti-inflammatory role.

We agree with the reviewer and tried (over the course of the last year) to do the conditional knock out of IFN γ receptor in OPCs using the commercially available floxed IFN γ rec mice crossed with the PDGFR Cre line. Unfortunately, recombination efficiency was unacceptably low and led to results that were not interpretable. We are addressing this point on a long-term basis by trying to develop new tools, but we are not able to conduct this specific experiment at this time. We do show specific data on the role of gamma interferon in the CNS, and we do intend to further analyze this point in the future.

Reviewer #3, expert on antigen presentation (Remarks to the Author):

Strength:

The study is nicely performed and carefully presented. The results are clear. In particular the data establishing the activation of the MHCI cross-presentation by IFN γ stimulation OPCs are convincing. In sum, this reports a new finding which may have an important significance in MS and brain infections. This highlight the possible role of OPC in local antigen cross presentation by MHCI.

Thank you.

Weakness:

*The immunopathological relevance of cross-presentation by MHCI for immunopathology is not assessed. Indeed, Th17-dependent immunopathology is largely dependent on the GM-CSF released by T cells which in turn activate monocytes to differentiate into inflammatory macrophages mediating tissue damage (Codarri L, Gyölvézi G, Tosevski V, Hesske L, Fontana A, Magnenat L, Suter T, Becher B. *Nat Immunol.* 2011 Jun;12(6):560-7; Spath S, Komuczki J, Hermann M, Pelczar P, Mair F, Schreiner B, Becher B. *Immunity.* 2017 Feb 21;46(2):245-260., Croxford AL, Lanzinger M, Hartmann FJ, Schreiner B, Mair F, Pelczar P, Clausen BE, Jung S, Greter M, Becher B. *Immunity.* 2015 Sep 15;43(3):502-14. Codarri L, Gyölvézi G, Tosevski V, Hesske L, Fontana A, Magnenat L, Suter T, Becher B. *Nat Immunol.* 2011 Jun;12(6):560-7).*

Thank for these comments and suggestions. We now discuss and reference these important papers. We appreciate the point that in vivo demonstration of pathologic cross presentation by OPCs would enhance the study. The definitive experiment to address the role of cross presentation of OPCs in vivo is to use a conditional knockout. Please see response to reviewer 2 and comments below.

The role of CD8+ T cells-dependent elimination of PCs via cross presentation is not incompatible with this view but would deserve some functional validation. For example, evidencing that CD8+ T cells impact on OPCs upon adoptive transfer (MOG97_114) by depleting CD8+ T cells with antibodies. It is anticipated that depletion of CD8+ T cell should increase the population of Fas+Kk+ OPCs and reduce the onset of caspase3/7 activation.

Towards this end, we have undertaken a number of in vivo studies over the past months but have encountered technical limitations. Wholesale pre-priming depletion of CD8+ T cells introduces a variable in the EAE model that would confound our studies on OPC survival. Depletion of CD8+ T cells prior to initiation of EAE has led to conflicting reports on the development and severity of EAE with some studies showing amelioration of EAE severity and some showing worsening by increasing the number of effector CD4s (reviewed in *Human Immunology* (2008) 69, 797-804). Nonetheless, we performed IP injections

starting 2 days before the removal of CPZ and CD4⁺ T-cell AT. Injections were continued once on the day of AT and then continuing either every other day or every 4 days using a dose range of between 200-400 µg/mouse. The protocol we used is consistent with standard experimental design of peripheral CD8 α depletion (*Eur J Immunol.* 2014 July ; 44(7): 1956–1966). We tried different regimens of depletion, but failed to obtain effective depletion of CD8⁺ T cells in the CNS and were thus unable to address this point despite multiple extensive attempts.

One other potential model we attempted to use was conducting the AT experiment in WT and OT-1 mice. The hypothesis of these experiments was that OT-1 recipient mice would have significantly reduced OPC caspase activity and death since the CD8 repertoire would theoretically be non-specific to self-antigens such as myelin. Not only have we had troubles with inconsistent induction of disease after AT of 2D2 T cells in these mice, but we learned of a report showing that OT-1 CD8s have been shown to cross-react with MOG peptide loaded into the MHC class 1 cleft *in vivo* (*Journal of Neuroscience* 25 March 2015, 35 (12) 4837-4850). The ability of OT-1 CD8 cells to cross react with myelin antigen eliminates the value of these mice to interrogate the desired experimental outcomes of caspase activity and death by cytotoxicity of the OPC population.

While the following data do not directly address the previous point, we have attempted to investigate the role of OPCs in the inflammatory response and include new data as follows: We stained for Granzyme B in non-lymphoid cells to examine if there was evidence for cellular cytotoxicity. We now show that OPCs make up a significantly greater proportion of the total granzyme B positive cells than microglia and dendritic cells (fig 7e-h), indicating that they are an active participant in this inflammation. Furthermore, analyses by flow cytometry for activated caspase and Live/Dead staining differences also show significant increases only in OPCs further highlighting the conclusion that OPCs are more vulnerable to granzyme B mediated cytotoxicity. While we show cross-presentation and cytotoxic killing *in vitro*, the extension of these specifically *in vivo* is supported by, but not definitively shown, by our new data, and we have significantly modified the text to reflect this distinction.

The immunopathological contribution of IFN γ signalling in OPCs is not tested in vivo. The study would be strengthened by providing a genetic evidence in vivo. This could be achieved by showing that conditional inactivation of IFN γ R or STAT1 in OPCs confers a significant protection from demyelination. Or minimally to evidence that IFN γ antibody blockade prevents the OPC activation and their elimination by CD8⁺ T cells in the context of adoptive transfer of Th17 CD4⁺ T cells.

Please see our response to reviewer 2. Very recently, the Yokoyama lab at Washington University published the first generation of beta-2 microglobulin floxed mice, which will allow conditional deletion of MHC I on OPCs, but these experiments will take more than a year to complete and we feel are beyond the scope of the present report. Although we feel that our *in vivo* data (in three different model systems) are compelling, we have modified

the discussion to reflect the shortcomings raised herein and have softened any definitive *in vivo* mechanistic claims. Accordingly, we removed cross-presentation and cytotoxicity from the title and clarify in the abstract that these were only shown *in vitro*.

Specific comments:

Figure 5 : the authors show that inhibitors of cross-presentation do not inhibit OT1 activation by OPC: this should be tested by titrating the SIINFEKL peptide across a broad range of concentrations.

We agree that inhibiting cross presentation is a critical point to expand upon. We conducted additional experiments in titrating both peptide and whole protein to determine the threshold and limits of antigen concentration in stimulating a response and show these titrations now in figure 6j. In order to test multiple inhibitors over multiple time points, we selected a concentration of ovalbumin that yielded a robust killing. We focused on whole protein in order to include the importance of processing. This figure now shows that even in a setting in which the caspase 3/7 is high, the inhibitors produce a significant effect, demonstrating that inhibition of the cross presentation pathway inhibits the ability of OPCs to activate T cells.

Figure 6 : The authors show that Fas expression is induced by IFN γ on OPCs but its contribution to cell death of OPC is not tested.

Thank you for this suggestion. We now show that the decoy receptor R3 (DcR3-Fc), which is known to inhibit FasL-mediated apoptosis, inhibits OPC death as does a GrB inhibitor thereby further implicating these pathways in the observed OPC death. Included in this assay was Q-VD-OPH a pan caspase inhibitor that directly blocks the activity of the reporter and apoptosis.

Figure 6 : The number of remaining live ODC should be monitored to directly assess cell death induced by cross presentation

We now provide the requested data. Our new data provide this direct quantification using the Incucyte live cell imaging system. We show the decrease of live OPCs after exposure to antigen and co-cultured with CD8⁺ T cells. The control OPCs with no peptide/CD8⁺s survive, while those with peptide and CD8⁺ T cells decrease over time. The quantification of this finding is now in figure 6c.

Figure 6 : do inhibitors of cross presentation (clhrq, ONX, CatS, cf fig. 5) inhibit caspase3/7 activation and cell death induced by OTI?

Yes, we now show that the inhibitors of cross presentation delay and inhibit caspase 3/7 activation and cell death significantly as determined by area under the curve analysis followed by one-way ANOVA using multiple comparisons. These data are now shown in figure 6i/j.

Figure 7d : the experiments should be better controlled using an isotope-matched control for the staining of Kk79-87 in all 4 conditions (+/-AT, +/-CPZ).

We agree that it is important to show specificity in staining and have included controls in each figure for non-specific staining. All our experiments were conducted with Fc receptor blockade and using cellular negative controls, which we think provide a higher level of specificity than an isotype, which may have unrelated background fluorescence. For example, in figure 7d, the experiment shows staining with Kk79-87, gated on OPCs, so that the same cells are used for analysis. When no antigen is present, there is no staining, and when antigen is present, there is staining for loaded cells. The absence of staining on the same cells that do not have any antigen present from the transfer serves as a negative specific cellular control. We have also included the widely accepted approach in flow cytometry of establishing a gate using florescence minus one (FMO) Ab.

REVIEWERS' COMMENTS:

Reviewer #1 (Remarks to the Author):

The authors have perfectly addressed the minor issues raised during the initial review of this manuscript. In addition, the authors have completely addressed initial concern surrounding quantitative analysis of apoptosis induction in OPCs. The work described in this manuscript is timely and of significant importance. It is the opinion of this reviewer that this manuscript should be published in Nature Communications.

Reviewer #2 (Remarks to the Author):

The authors thoroughly addressed the reviewers' concerns by correcting mistakes/omissions in the manuscript and figures/legends, including new data, and explaining reasons for not conducting some of the experiments suggested by the reviewers. The manuscript has been substantially improved and this reviewer does not have additional comments.

Reviewer #3 (Remarks to the Author):

In the revised version of the ms, the authors have convincingly addressed my previous comments on :

- SIINFEKL titrations
- the role of FasL
- methods for the evaluation fo cell death
- role XP inhibitors on apoptosis
- antibody staining controls

Unfortunately, attempts to establish the physiological role of the process with PDGFR^{cre}IFN γ FL/fl or antibody-mediated CD8⁺ T cell depletion were proven unsuccessful for technical reasons.

These attempts did not led to conclusive results and do not contradict the model proposed.

In its current form the ms still brings novelty and highlight an important effector mechanisms underlying EAE immunopathology whose physiological relevance (beyond adoptive transfer experiments) will have to be further investigated.